



# Biological response to hydrodynamic factors in estuarine-
# coastal systems: a numerical analysis in a micro-tidal bay.
Marta F-Pedrera Balsells [1]*, Manel Grifoll[1], Margarita Fernández-Tejedor[2], Manuel
Espino[1], Marc Mestres[1], Agustín Sánchez-Arcilla[1].
[1] Maritime Engineering Laboratory (LIM), Catalonia University of Technology (UPC), 08034
Barcelona, Spain;
[2] Institute of Agriculture and Food Research and Technology (IRTA), Crtra. Poble Nou, s/n Km 5,5.
43540 Sant Carles de la Rapita, Spain;
* Correspondence: marta.balsells@upc.edu (M.F-P.B.)
**Abstract:**
Phytoplankton primary production in coastal bays and estuaries is influenced by multiple physical
variables, such as wind, tides, freshwater inputs or light availability. In a short-term perspective these
factors may influence the composition of biological variables such as phytoplankton biomass, as well as
the amount of nutrients within the waterbody. Observations in Fangar Bay, a small, shallow, stratified and
micro-tidal bay in the Ebro Delta (NW Mediterranean Sea), have shown that during wind episodes the
biological variables undergo sudden variations in terms of concentration and distribution within the bay.
The Regional Ocean Model System (ROMS) coupled with a nitrogen-based nutrient, phytoplankton,
zooplankton, and detritus (NPZD) model has been applied to understand this spatio-temporal variability of
phytoplankton biomass in Fangar Bay. Idealised simulations prove that during weak wind events ($< 6$ m·s$^{-1}$
), the stratification is maintained and therefore there is not dynamic connection between surface and bottom
layers, penalizing phytoplankton growth in the whole water column. Conversely, during intense wind
events ($> 10$ m·s$^{-1}$) water column mixing occurs, homogenising the concentration of nutrients throughout
the column, and increasing phytoplankton biomass in the bottom layers. In addition, shifts in the wind
direction generate different phytoplankton biomass distributions within the bay, in accordance with the
dispersion of freshwater plumes from existing irrigation canals. Thus, the numerical results prove the
influence of the freshwater plume evolution on the phytoplankton biomass distribution, which is consistent
with remote sensing observations. The complexity of the wind-driven circulation due to the bathymetric
characteristics and the modulation of the stratification implies that the phytoplankton biomass differs
depending on the prevailing wind direction, leading to sharp Chl $a$ gradients and complex patterns.
**Keywords:** phytoplankton biomass, ROMS-NPZD model, wind, biological parameters, physical
parameters, Fangar Bay.
## 1. Introduction
The intense biological activity of estuaries and coastal bays and their importance as a source of resources
and socio-economic services is well known. The use of these areas as aquaculture zones has provided great
benefits as well as great problems (Ramón et al., 2007; Llebot et al., 2010, 2011; Soriano-González et al.,
2019). The influence of the terrestrial environment and human activity in these domains provide the
nutrients necessary to create ecosystemic value (Lohrenz et al., 1997). The biological evolution of these
waterbodies is strongly affected by physical factors. For instance, strong winds may controls the inner water
circulation (Geyer, 1997; Alekseenko et al., 2013; Cerralbo et al., 2015) and, together with topographic
effects, can even cause the current to flow against the wind direction in the central channels (Xie & Li,
2018; F-Pedrera Balsells et al., 2020a). Freshwater inputs can also have a considerable effect on the water
circulation (Cerralbo et al., 2014) and, acting as fluvial nutrient suppliers, can determine the temporal and
spatial variability of phytoplankton biomass (Geyer et al., 2018; Jiang et al., 2020). In this sense, the use of
coupled physico-biological numerical models as a tool to understand the complexity of the phytoplankton
regulatory mechanism in estuaries has increased in recent years, complementing *in situ* data and satellite
imagery (Llebot et al., 2010; Artigas et al., 2014; Jiang et al., 2020). These numerical models can provide
information on the current state of the estuary, create hypotheses and numerical experimentation, and
predict certain events and ecosystem responses (Stow et al., 2009; Llebot et al., 2010).



In small and shallow estuaries, the effects of both the physical and biological mechanisms become more complex due to the geometry of the basin itself. Moreover, in the bays of the Ebro Delta (i.e. Alfacs in the southern hemidelta and Fangar in the northern hemidelta), freshwater discharges from rice field irrigation channels also play an important role, together with aquifer contributions, being an important source of nutrients for the coastal areas (Jou et al., 2019), both inorganic and organic. In a small-scale coastal bay such as Fangar Bay, where depths are only of a few metres, chlorophyll a (Chl $a$) concentrations tend to show a high variability on a seasonal scale rather than on an interannual scale (Llebot et al. 2011), with higher concentrations found during the summer (Ramón, Fernández, and Galimany 2007). In this sense, Chl $a$ concentrations in Fangar Bay tend to show a larger variability as compared to other coastal domains such as the Ría de Arousa (Ramón, Fernández, and Galimany 2007) or Alfacs Bay (Artigas et al. 2014).

Previous investigations in Fangar Bay revealed that, during wind episodes, the biological variables undergo sudden variations in terms of concentration and distribution within the bay (F-Pedrera Balsells et al., 2021). *In situ* measurements obtained during specific field campaigns and remote sensing observations suggested a link between the breaking of the stratification during these episodes and the Chl $a$ distribution in the bay, with intense winds causing an increase in the Chl $a$ concentration values. The role of the discharges from the irrigation canals remained unclear because the freshwater outflow was constant during the field campaigns. However, the spatial-temporal variability of the Chl $a$ concentration observed in the field campaigns was quite complex due to the many factors involved, and deserved additional modelling efforts. The purpose of the present study is to further this first assessment by investigating the biological response of the bay to wind episodes and freshwater inputs through the combination of idealised numerical simulations and observations. For this, a biological model is embedded into a validated hydrodynamic model to reproduce the dynamics within the bay. Extensive field data and previous hydrodynamic knowledge converts Fangar Bay in a unique study area to investigate the biological response in an area with large spatio-temporal variability on Chl $a$ evolution.

The paper is organised as follows: Section 2 presents a detailed description of the study area and the model used, as well as a brief description of the pre-processing satellite images. Section 3 presents the results of the numerical simulations concerning the evolution phytoplankton, salinity and nutrients, as well as a comparison with satellite images. Section 4 discusses the effects of the wind and freshwater plume on the biological variables comparing with sites. Finally, section 5 summarises the main conclusions and suggests future works to be carried out.

## 2. Material and Methods

2.1. Study area

Fangar Bay is part of the Ebro Delta (NW Mediterranean Sea), which reaches about 25 kilometres offshore and forms two semi-enclosed bays, Fangar to the north and Alfacs to the south. Of these, Fangar Bay is the smallest, extending over 12 km$^2$, with a length of about 6 km, a maximum width of 2 km and a volume of water of 16x10$^6$ m$^3$ (Delgado and Camp 1987). The average depth is 2 m, with a maximum of 4 m (see bathymetry in Figure 1). Its connection with the open sea is oriented to the NW, and is approximately 1 km wide (Garcia and Ballester 1984), although it is currently narrowing because of the accumulation of sediment from the beach located to the north (Archetti, Bernia, and Salvà-Catarineu 2010).

The wind regime in the Fangar Bay area is characterized by the presence of S/SE sea breezes – which do not exceed 6 m·s$^{-1}$ during spring and summer– and strong winds from the N and NW of more than 12 m·s$^{-1}$ in autumn and winter (Bolaños et al. 2009; Grifoll et al. 2016). The most frequent wind throughout the year is locally known as Mistral, which is characterized by strong gusts of cold and dry wind from the NW (Garcia and Ballester 1984). These winds are associated to the general weather pattern and occur throughout the year, but show maximal strength and persistence during the colder months. Additionally, E and NE winds that can also be quite intense (~10 m·s$^{-1}$) are responsible for local rain events and transient increases of the local mean sea level at the coast (Muñoz 1990).

Both the Ebro Delta bays receive freshwater inputs from the canals irrigating the Delta paddies. This
freshwater outflow is regulated by the rice cultivation cycle throughout the year. In Fangar Bay, the canals
are open between April and November, discharging a mean flow of 7.23 m³·s⁻¹ (*SAICA Project, 2013.*
*Available online: https://www.saica.co.za/ (accessed on 30 January 2020)*), whereas the outflow is
negligible from December to March, when the channels are closed (Perez & Camp, 1986). There are two
main freshwater discharges in Fangar Bay: one in the Illa de Mar harbour inside the bay (IM in Figure 1)
and the other one, Bassa de les Olles, located at the bay mouth (BO in Figure 1). In addition to these,
freshwater inputs are also expected inside the bay from groundwater sources (Camp and Delgado 1987),
and along the coastline where freshwater inflows regulated by gravity according to the sea level occur. In
both cases, the expected freshwater inflow is smaller than that discharged from the regulated irrigation
channels.



**Figure 1. Location of the study area. The red circles show the two main points of freshwater discharges
(Bassa de les Olles (BO) and Illa de Mar (IM). The yellow stars show the location of the control points used
for the numerical model results. The bathymetry is also shown in the figure.**

Fangar Bay is micro tidal, with a tidal range smaller than 1 m, which accentuates the action of the wind,
and is stratified most of the year mainly due to the freshwater flows rather than to the atmospheric heat
fluxes. Because of its bathymetry and complex geometry there is a strong transverse variability of the water
flows, particularly for prevalent up-bay wind episodes (NW winds), during which up-bay flow occurs in
the lateral shoals and down-bay flow in the central channel for up-bay wind pulses. These winds also cause
homogenisation of the whole water column. On the other hand, during calm periods the water circulation
is complex: current velocities are very small and lack a clear pattern, and the bay is strongly stratified due
to the freshwater inputs from the drainage channels (F-Pedrera Balsells et al., 2020a).
2.2. Numerical model and experiments design
A set of numerical experiments were conducted using the Regional Ocean Model System (ROMS) to
analyse the link between the hydrodynamic and Chl *a* response to the wind in small and shallow estuaries.
The ROMS numerical model is a 3D, free-surface, terrain-following numerical model that solves the
Reynolds-Averaged Navier-Stokes equations using hydrostatic and Boussinesq assumptions (Shchepetkin
and McWilliams 2005). ROMS uses the Arakawa-C differentiation scheme to discretize the horizontal grid
in curvilinear orthogonal coordinates and finite difference approximations on vertical stretched coordinates
(Haidvogel et al. 2007). The numerical details of ROMS are described extensively in (Shchepetkin and
McWilliams 2005). This model has been used and validated in similar bays and estuaries, such as Alfacs





Bay located south of the Ebro Delta (e.g. (Cerralbo et al., 2014, 2015, 2019)) and in the Fangar Bay (see
Annex 1). The domain used for the experiments consists of a regular 107x147 grid with a horizontal
resolution of about 70 m and 10 sigma levels in the vertical direction. The model boundary is located 10
points away of the mouth entrance to avoid boundary noise. The hydrodynamic bottom boundary layer was
parametrised with a logarithmic profile using a characteristic bottom roughness height of 0.2 m. The
turbulence closure scheme for the vertical mixing was the generic length scale (GLS) tuned to behave as a
k-ε (Warner et al., 2005). Horizontal harmonic mixing of momentum was defined with constant values of
$5\ m^2\cdot s^{-1}$.
The NPZD numerical model coupled with ROMS model includes dissolved inorganic nitrogen,
phytoplankton, zooplankton and detritus (Franks 2002). The initial nitrate concentration was taken from
field data collected by IRTA between the years 2009-2012 (ACA, 2012), and the initial phytoplankton
concentrations were collected from observation data during the year 2019, whereas the initial zooplankton
concentration was estimated from the literature (Rico 2015; Powell et al. 2006). The units in which these
data were collected were $mg\cdot m^{-3}$. The NPZD model uses $mmol\cdot m^{-3}$ units, so a conversion has been made
using the mole fraction of Chl *a* (see Table 1). The rest of input variables for the ROMS-NPZD model were
acquired from Llebot et al. (2010), and are detailed in Appendix B (Table B1), together with the model
equations. Short-term simulations (5 days each) were carried out to analyse the response of biological
variables to the wind. This simulation length exemplifies the typical wind events in the area, lasting from
3 to 5 days (except the daily sea breeze during spring and summer). Six experiments were designed, varying
the wind direction and intensity, as well as the freshwater from the channels. The wind parameters are based
on wind measurements in the Fangar area (F-Pedrera Balsells et al., 2020a), with weaker down-bay winds
(associated to daily sea breeze, DW6 simulation), NW up-bay winds (UW10 simulation), strong NW up-
bay winds (UW12 simulation) and SE down-bay winds (DW8 simulation). For theoretical comparison, a
simulation was also carried out with $0\ m\cdot s^{-1}$ wind intensity (CALM simulation). All simulations are
summarized in Table 1. Temperature and salinity conditions were in accordance with those measured within
the bay (see field campaign description in F-Pedrera Balsells et al., 2021). Freshwater inputs were activated
to monitorize the evolution of nutrient inputs from the irrigation channels. Both channels (BO and IM,
Figure 1) provides nutrients that will be presumably dispersed within the bay due to the combined action
of currents and wind.
**Table 1. Summary of the idealized numerical simulations using the ROMS-NPZD model for Fangar Bay.**

| Simulation | Wind direction | Intensity wind ($m\cdot s^{-1}$) | Channel flow ($m^3\cdot s^{-1}$ each channel) | Initial nitrate concentration ($mmol\cdot m^{-3}$) | Initial phytoplankton biomass ($mmol\cdot m^{-3}$) | Initial zooplankton biomass ($mmol\cdot m^{-3}$) |
|---|---|---|---|---|---|---|
| **CALM** | - | 0 | 7.5 | 2.73 | 0.27 | 0.08 |
| **DW6** | Down-bay wind | 6 | 7.5 | 2.73 | 0.27 | 0.08 |
| **UW10** | Up-bay wind | 10 | 7.5 | 2.73 | 0.27 | 0.08 |
| **DW8** | Down-bay wind | 8 | 7.5 | 2.73 | 0.27 | 0.08 |
| **UW12** | Up-bay wind | 12 | 7.5 | 2.73 | 0.27 | 0.08 |
| **UW12fr** | Up-bay wind | 12 | 3 | 2.73 | 0.27 | 0.08 |






2.3. Satellite image processing

To qualitatively compare the numerical modelling results with real observations, satellite images from Sentinel-2, level 1-C, are used. These satellites carry a single optical instrument, the MultiSpectral Imager (MSI), and its swath width (290 km) and high revisit time (10 days at the equator with one satellite and 2–3 days at mid-latitudes) support monitoring of Earth's surface changes. Chlorophyll $a$ concentrations were computed automatically by the Sentinel Application Platform (SNAP) (https://step.esa.int/main/toolboxes/snap/, accessed on 25 February 2021). The MSI sensor has had an atmospheric correction applied to it with a C2RCC processor (Case 2 Regional CoastColour, Brockmann et al., 2016) to obtain the Chl $a$ images. The images correspond to remote sensing obtained after intense wind episodes (see details in F-Pedrera Balsells et al., 2021).

**3. Results**

Four points within the bay have been chosen to investigate the temporal evolution of the biological variables obtained from the NPZD model: in the mouth area (M1), in the centre of the bay (M2), in a coastal area in front of the IM discharge point (M3) and in the innermost part of the bay (M4). Both channels (BO and IM, Figure 1) provide nutrients which increase the concentration of phytoplankton biomass within the bay. Figure 2 shows the time series of the numerical simulation in terms of nitrates and phytoplankton at the four control points. The nitrate concentration tends to decrease gently during the simulation, consistently with the increase in phytoplankton biomass and zooplankton. All simulations show larger concentrations of phytoplankton biomass at the surface due to the freshwater fluxes, as will be discussed later. Substantial differences of phytoplankton biomass between surface and bottom layers are evident in M1, where the stratification tends to be larger in contrast to the shallowest point (M4). The inner point M4 also shows a clear correlation of the wind intensity and the phytoplankton biomass values: as the up-bay wind intensity increases (i.e. UW12, larger than UW10) the phytoplankton biomass also increases. In all cases the numerical simulations suggest large temporal and spatial variability within the bay.



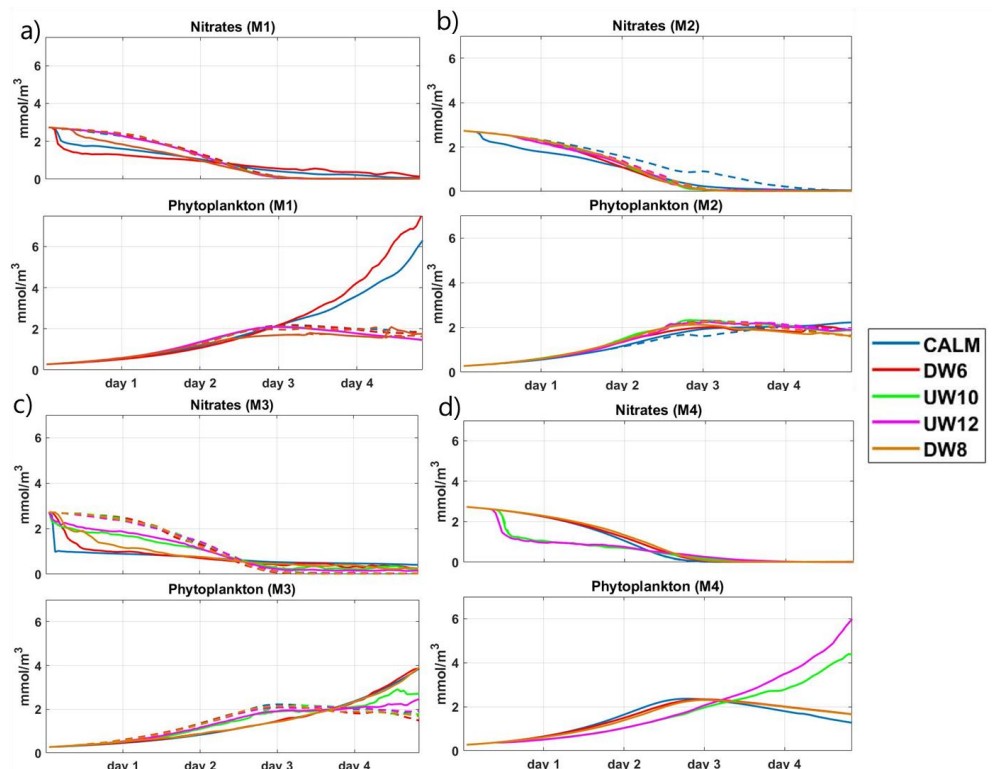

**Figure 2. Time series of the nitrates and phytoplankton biomass at different points of the bay: (a) M1, (b) M2, (c) M3 and (d) M4. The different colours show the different simulations with in function of the wind. Solid line shows surface numerical results, dashed line shows bottom numerical results.**

Intense wind was associated to the homogenisation of the initially stratified water column for strong
wind episodes. In particular, for the UW10 and UW12 simulations (moderate and strong up-bay wind),
both surface and bottom phytoplankton time series coincide at all control points. Figure 3 shows the vertical
profiles of phytoplankton biomass and salinity after three days simulation at the four points mentioned
above for the DW6 (weak down-bay wind), DW8 (SE down-bay wind) and UW12 (NW up-bay wind)
simulations. These profiles show homogeneous concentrations of phytoplankton biomass and salinity in
the water column after strong wind episodes (DW8 and UW12). In contrast, during weaker winds (DW6)
the saline stratification tends to remain in M1 and M3, leading to larger presence of phytoplankton biomass
in the surface layers. At the innermost point of the bay (i.e., M4), the phytoplankton biomass is
homogeneous in all simulations since, due to the shallowness of the area (lesser than 1 m deep), even weaker
winds are able to mix the water column. The comparison of profiles at M3 shows a high variability of
phytoplankton biomass values: DW8 shows larger values of phytoplankton biomass as compared to DW6.
This means that the mixing mechanism can favour the increase of phytoplankton biomass. Finally, the
comparison in M3 between DW8 and UW12 also suggests the effect of the freshwater plume on the
phytoplankton biomass, which will be discussed later.
Differences in growth rates between phytoplankton and zooplankton biomass are observed in the time
series. While phytoplankton can multiply rapidly under favourable concentrations of light and nutrients,
increases in zooplankton numbers often lag considerably behind due to their slower generation times.
Consequently, when phytoplankton biomass peaks and nutrients decline, zooplankton biomass may remain
low as they begin to grow in response to the high food supply (Figure not shown). These relations are
consistent with the diagram shown in the Appendix B (B1).

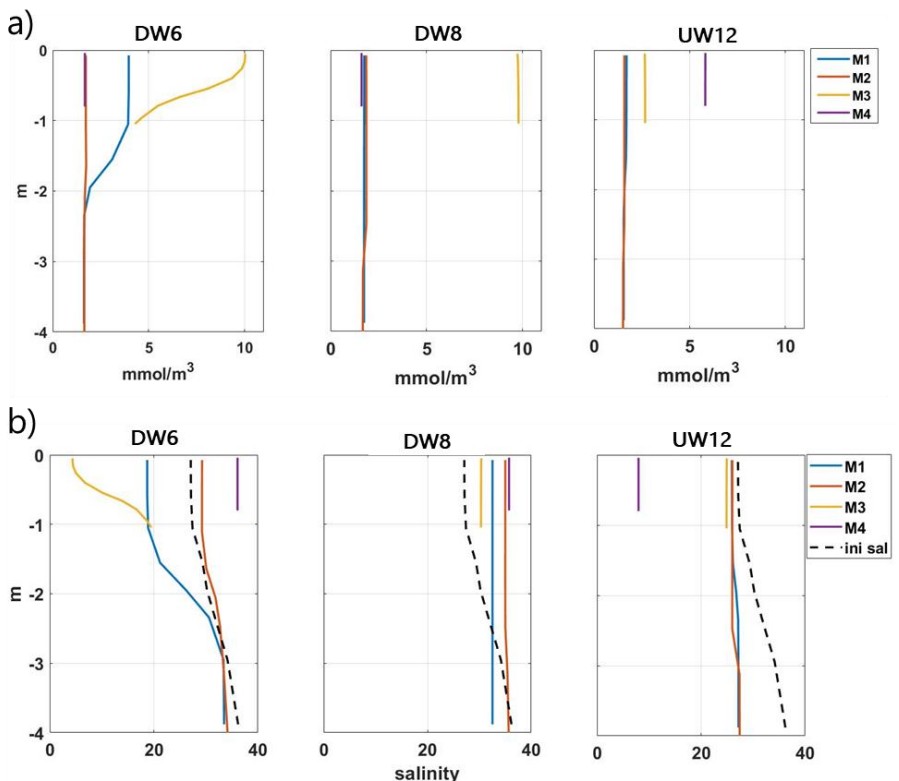

**Figure 3. Vertical Chl *a* profiles (a) and vertical salinity profiles (b) as a function of wind events simulations at the four sampling points: M1 (blue), M2 (orange), M3 (yellow) and M4 (purple). Dashed black line shows the initial salinity.**

Differences in the horizontal distribution of phytoplankton biomass are also observed in Figure 3a. During calm winds, the highest biomass value is found in front of the freshwater discharge points (M1 and M3). During strong NW up-bay winds (UW12), the highest biomass values are found in the innermost area of the bay (M4), while during strong SE down-bay winds (DW8) the largest biomass concentration is observed at the coastal point inside the bay (i.e., M3), with higher values than during the DW12 simulation.

In order to examine the spatial variability, Figure 4a shows the differences in phytoplankton biomass at the end of the simulation in comparison to the initial concentration values. Surface and bottom values are displayed according to the terrain following sigma coordinates of the numerical model. All the simulations present positive values, indicating an increase of phytoplankton biomass due to the nutrients provided by the freshwater. During up-bay winds (simulations UW12 and UW12fr), the phytoplankton biomass increases in the inner zone, both at the surface and at the bottom, with concentrations larger than 5 mmol·m$^{-3}$. This coincides with vertical mixing of the water column, as shown by the salinity distributions (Figure 4b). In contrast, during no wind and weak (DW6) winds, the highest phytoplankton biomass (4 mmol·m$^{-3}$) is located in front of the discharge points, with the largest values obtained at the point inside the bay (M3, 10 mmol·m$^{-3}$ at the surface). For no wind simulation (CALM simulation) stratified conditions remain. In this case, bottom concentrations are small (1 mmol·m$^{-3}$) in comparison to other cases, highlighting the positive effect of strong winds on the vertical distribution of phytoplankton biomass. During the DW8 simulation, the highest concentrations are also observed near the discharge points (M1 and M3), with the highest values found towards the mouth, consistent with the presence of a low-salinity plume. Overall, there is a correspondence between the freshwater plume and the phytoplankton biomass. Therefore, the wind-driven evolution of the plume has a very important impact on the final distribution of



Chl *a*. In the same way, it can be seen in Figure 3 that winds of similar intensities but different directions
(DW8 vs. UW10) lead to very different results in terms of the horizontal distribution of phytoplankton
biomass. Finally, the results of an additional simulation similar to UW12 but with half the canals' outflow
rates (UW12fr) revealed a horizontal distribution similar to UW12, but with smaller phytoplankton biomass
concentrations associated to a lower nutrient input.

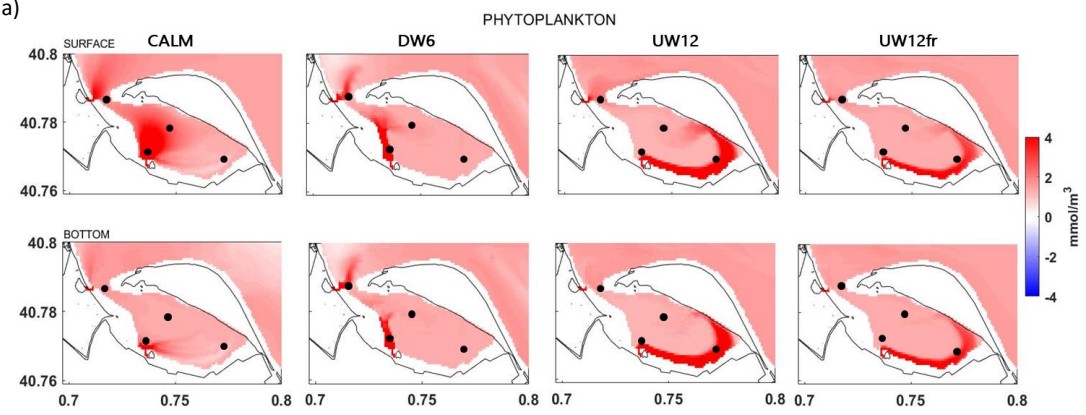

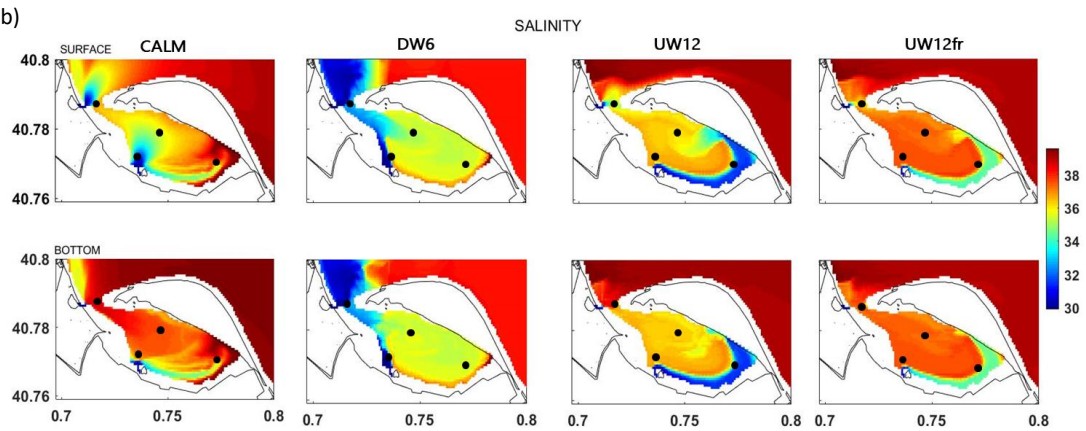

**Figure 4. Differences in phytoplankton biomass (surface and bottom) at the end of the simulation in comparison to the to the initial concentration values (a) and surface and bottom salinity (b) according to different numerical simulations. The numerical control points are also marked with black dots.**

Figure 5 shows a comparison between the model results and the Sentinel-2 satellite images, in periods
of calm or weak wind, and strong wind which produces mixing (NW up-bay wind). The satellite images
correspond to 15-Jul-2019 (Figure 5d) and 11-Aug-2019 (Figure 5c), one day after an up-bay wind episode
occurred on 14-Jul-2019, and during a sea-breeze period (weak winds) between 30 July and 12 August,
respectively. Note that the satellite image of the up-bay wind episode is from a few days after the wind has
blown, while the model results correspond to the blowing of a steady wind. In spite of this, there is an
identifiable correlation between model and images. For calm or weak down-bay wind (sea breeze), the
phytoplankton biomass is relatively low, only present in the areas close to the discharge channels following
the coastline consistent with the wind-driven currents due to sea breeze. During NW up-bay winds,
phytoplankton biomass increases in the inner zone and is later dispersed within the bay. In any case, it
should also be taken into account that the satellite, being such a shallow area, does not only show data on



phytoplankton chlorophyll but also on macrophytes, which are very present in this bay (Soriano-González
et al. 2019).

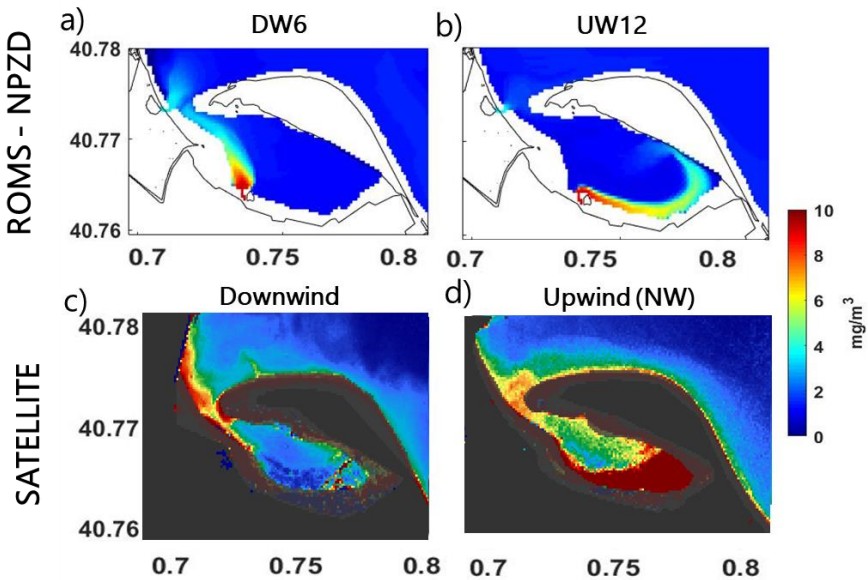

**Figure 5. Comparison of surface model results with Sentinel 2 satellite images. (a) and (b) show model results during breezes and strong NW winds, respectively. (c) and (d) show corresponding satellite images with the same wind episodes.**

**4. Discussion**
The phytoplankton distribution is controlled by turbulent mixing and advection factors, which are
affected by physical forcings such as wind, tides and continental freshwater input. In tidally dominated
estuaries or upwelling areas, phytoplankton biomass is distributed according to spring blooms, where algal
blooms generated during upwelling events are transported to the bays through various physical mechanisms
(tidal stirring, and gravitational and wind-driven circulation (Hickey and Banas, 2003; Martin et al., 2007),
as occurs in the Eastern Scheldt Bay (Jiang et al. 2020) or the Rías Baixas of Galicia (Reguera et al., 1993).
On the other hand, in estuaries where the tide is practically non-existent and the depth is small,
phytoplankton growth is limited by nutrients and the turbidity of the water due to large inputs of sediment
from rivers and channels, such as in the Chilika lagoon (Srichandan et al., 2015) or our study area, the
Fangar Bay.
From a hydrodynamic point of view Fangar Bay is complex due to its shallowness and intricate
bathymetry (Llebot, 2010; F-Pedrera Balsells et al., 2020a). Llebot et al. (2010) implemented a numerical
model in Fangar Bay to determine the temporal distribution of phytoplankton and nutrients throughout the
year. They determined that the highest concentration of phytoplankton occurred during the first months of
the year with open irrigation channels, i.e., in the spring and early summer, but the role of episodic wind
events remained unclear. The present study is centred on the phytoplankton distribution resulting from
different short-term wind episodes typical of the Ebro Delta area, and focuses on the summer months for
which the bio-hydrodynamics are well defined (F-Pedrera Balsells et al., 2021). The numerical results have
proved that the wind affects the direction and magnitude of surface currents, disperses or reinforces fronts,
and induces vertical mixing, in accordance with other investigations (Geyer, 1997; Llebot et al., 2014; F-
Pedrera Balsells et al., 2020a). The distribution of phytoplankton biomass in Fangar Bay agrees with water



currents driven by the local winds and modulated by the complex bathymetry of the basin (F-Pedrera Balsells et al., 2020a) and the evolution of the freshwater plume of the drainage channels (Figure 4b).

The model results have shown the combined effect of the wind on phytoplankton biomass distribution within the bay from two perspectives. On the one hand, intense wind episodes are able to break the stratification, mixing the water column and leading to an increase of phytoplankton biomass in the deeper levels of the bay. Simulations with no wind (CALM), in which the bay remains stratified, are characterized by the presence of a physical barrier that prevents nutrient vertical transfer, so the phytoplankton biomass remains in the surface layers. This case shows how the absence of wind (or even the presence of sea breezes) causes the phytoplankton biomass dispersion to be governed by the estuarine circulation of the bay, with phytoplankton biomass decreasing in the seaward direction. A similar seaward negative gradient of phytoplankton is found in other estuaries and coastal systems in which the nutrient gradients control the phytoplankton distribution (Soetaert et al., 2006; Gomez et al., 2018). When strong NW up-bay winds blow in Fangar Bay, the water column homogenises, making nutrients available throughout the column, both at the surface and at the bottom (i.e., UPW10 and UW12 simulations). The phytoplankton biomass is advected towards the inner part of the bay, following the water currents induced by the NW winds, not only at the surface but throughout the water column. With strong SE down-bay winds (DW8 simulation), phytoplankton biomass increases near the discharge channels and the phytoplankton biomass distribution follows the water circulation driven by SE winds: seaward flow in the lateral shoals (F-Pedrera Balsells et al., 2020a). The strong winds episodes suggest a non-uniformly distribution of phytoplankton biomass with irregular patterns and patches attributed to a dominant source factor (see examples in Ahel et al., 1996; Geyer et al., 2018; Jiang et al., 2020), which in the case of Fangar Bay is a role played by the discharge channels. In consequence, following the categorization exposed by Jiang et al. (2020) in terms of spatial patterns of phytoplankton biomass in estuaries and coastal bays, Fangar Bay may be included in different typologies depending on the wind configuration from a short-term perspective.

The second effect of the wind is related to the wind-driven plume dispersion. Freshwater discharges from the irrigation channels are the dominant driver of salinity and nutrient gradients. In Fangar Bay there is a colimitation of nitrogen and phosphorus, with the most limiting nutrient changing throughout the year, depending on the variability of sources and sinks of both nutrients (Llebot et al. 2010). The NPZD model considers nitrogen only and assumes no phosphorus limitation in phytoplankton growth. This sets a limit on the full understanding of Fangar Bay's dynamics, but our analysis provides a first interpretation of the data. Freshwater discharges also vary over the year, depending on whether these irrigation channels are closed (January to March), open (April to November) or semi-open (November and December). In F-Pedrera Balsells et al. (2021) it was observed that after strong NW up-bay wind episodes in the bay, phytoplankton biomass tends to increase within the bay, but it could not be determined whether this behaviour extended to the entire water column as the simulations presented herein suggest. As described in other works (Simpson & Bowers, 1981; Horsburgh et al., 2000) vertical density stratification is an important determinant of ecosystem characteristics.

Ultimately, as a small, shallow, micro-tidal bay, wind generates very complex currents and causes large spatial and temporal variability in the distribution of phytoplankton biomass. Chl *a* peaks usually form at the front of the river plume, either by rapid nutrient assimilation and growth or by aggregation along the strong salinity gradient of this transition (Geyer et al., 2018). This explains why the highest biomass levels can be found following the river plume, as shown in Figure 4. Therefore, the same simulations were performed by halving the channel outflow and the results on phytoplankton biomass distribution were the same, only lower, due to lower nutrient input. Some studies have shown that Chl *a* concentration is higher in the freshwater areas of the bay and decreases as salinity increases. This leads to high phytoplankton biomass in the plume formed by freshwater tributaries, which discharge high levels of nutrients, as can happen in the Scheldt River and Western Scheldt Estuary in Belgium (Soetaert et al., 2006).

Freshwater discharges from irrigation channels also control water residence times within the bay. Prolonged residence times generally facilitate the growth and accumulation of phytoplankton biomass (Wan et al. 2013). The location and magnitude of phytoplankton biomass can be partly explained by residence time, although phytoplankton productivity may be affected by other factors such as nutrient availability, light, temperature and zooplankton grazing (Wan et al. 2013). In Fangar Bay, residence time is in the range





of about 20 days in the middle zone and about 40 days in the innermost quasi-stagnant zone (F-Pedrera
Balsells et al., 2020b). This work also shows that an increase in freshwater discharge through the inner
channel (IM) helps to decrease the residence time in the innermost zone (F-Pedrera Balsells et al., 2020b).
In turn, therefore, a reduction in river discharge increases residence time and may allow a higher
concentration of phytoplankton to accumulate within the estuary. Our results show a higher concentration
of phytoplankton biomass in the innermost zone consistent with the larger residence time.
**5. Conclusions and future works**
Results based on *in situ* and remote observations and numerical models conclude that the biological
variables in small-scale, shallow and micro-tidal bays (such as Fangar Bay) shows strong gradients due to
the influence of the wind and the freshwater plume evolution. Strong winds have a double impact: i)
breaking down the stratification and mixing the water column, leading to an increase of phytoplankton
biomass at the bottom, and ii) distributing the canal-borne nutrients within the bay, resulting in an irregular
pattern of phytoplankton biomass. Due to the predominance of the wind forcing on the bay's water
circulation, different wind directions and/or intensities may have a completely different effect on
phytoplankton biomass distribution. In this sense, wind variability explains the complex pattern of
phytoplankton biomass observed in the *in situ* measurements and remote sensing, characterized by sharp
horizontal gradients. In particular, the link between the hydrodynamics and the phytoplankton evolution in
Fangar Bay can be summarized as shown in Figure 6. This figure is a conceptual diagram of the estuarine
processes affecting phytoplankton biomass distribution in a small-scale, micro-tidal bay. The distribution
of these nutrients is further influenced by the surface currents induced by the different winds in the area.
With weak down-bay winds (i.e. sea-breeze), stratification is maintained within the estuary, so higher
phytoplankton biomass near the discharge points and at the surface layer are found. During strong up-bay
and down-bay winds (i.e. North-westerlies, and South-easterlies, respectively), the stratification is broken,
so the nutrients discharged from the channels are distributed homogeneously throughout the water column,
facilitating phytoplankton growth in the deeper layers. This growth expands horizontally according to the
wind-driven currents: towards the inner zone through the lateral shoals, in the case of up-bay winds, or
towards the mouth zone, in the case of down-bay winds.

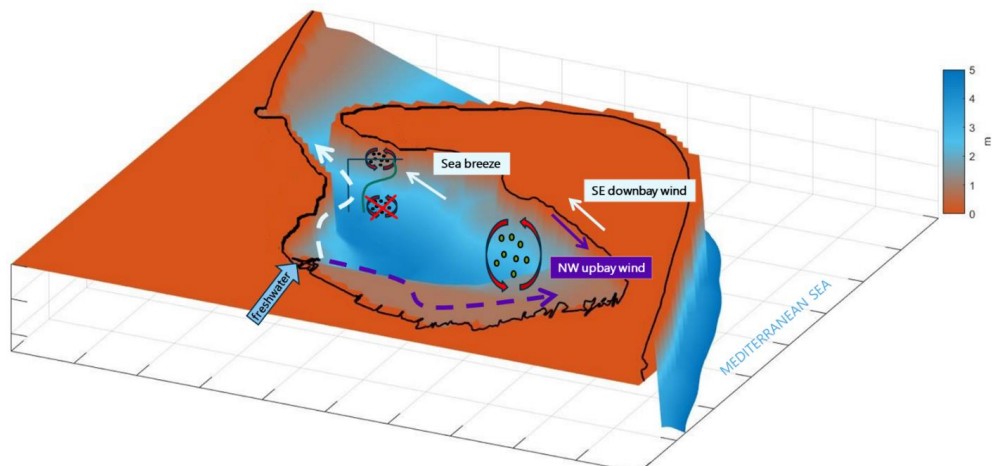

**Figure 6.** Conceptual diagram of estuarine processes affecting phytoplankton distribution in Fangar Bay. Strong NW up-bay winds cause mixing of the water column and the freshwater plume to move inland. Strong SE down-bay winds also cause mixing, but the freshwater plume moves seaward. Sea breezes also cause a seaward displacement of the freshwater plume, but does not break the vertical stratification, so there is a difference between phytoplankton biomass at the surface and at the bottom.

Fangar Bay is complex from both a hydrodynamic and biological standpoint. Different phytoplankton
patterns are identified depending on the meteorological conditions and, to account for this, different





idealised simulations were designed in order to approach each scenario separately. Even so, there are
processes that remain unexplored such as the resuspension of Chl *a* containing biomass or the effect of
long-duration wind episodes, as well as the change in limiting nutrients that often occurs in such
environments, affecting phytoplankton biomass, composition and seasonal cycling (D'Elia et al., 1986;
Fisher et al., 1992). The availability of N or P inside the bay also influences the biochemical composition
of phytoplankton (Estrada et al., 2008). These topics remain to be studied in future work, as does the
analysis of the impact of these dynamics on zooplankton and detritus, which are two variables also taken
into account by the ROMS-NPZD model. In any case, the combined analysis of observations and numerical
models has provided compelling results and opens new perspectives to understand the short term dynamics
of shallow and micro-tidal bays to meteorological events from a combined hydro-biological point of view.
**6. Appendix A.** Model validation.
The numerical implementation in Fangar Bay consists of a telescopic three-grid two-way nested ROMS
scheme, with a finer bay grid (resolution of about 23 m) embedded within a cascade of coarser grids (see
Figure A1). The model has been validated by comparing modelled surface velocities from the coastal
domain (~ 350 m) with HFR (High Frequency Radar) data, and modelled currents from the finest domain
(~ 23 m) with vertical current profiles measured inside Fangar Bay during an October-November, 2017
field campaign. The observational data were, amongst others, current velocity and direction obtained every
10 min in 25 cm thick layers distributed from the bottom to the surface.

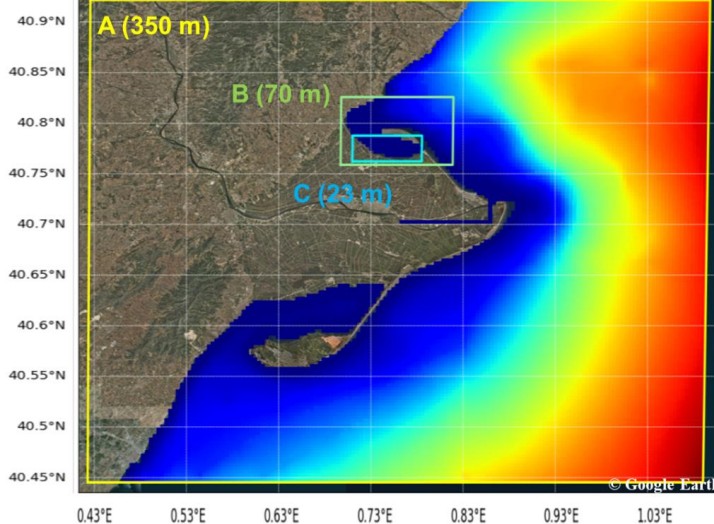

**Figure A1. Ebro Delta and the Fangar Bay, with the telescoping domains used in the system. Conditions for ROMS at the A domain are obtained either from CMEMS-IBI or CMEMS-MED. Map from Google Earth (Data SIO, NOAA, U. S. Navy, NGA, GEBCO, Image Landsat/Copernicus © 2018 Google.**

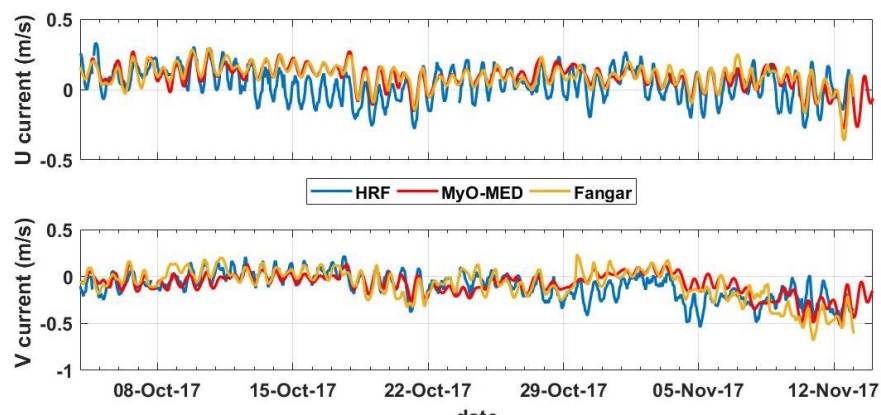

**Figure A2. Surface current components U (east-west) and V (north-south) measured by the HRF radar (blue line) and modelled my CMEMS-MED (red) and the Fangar nested suite (yellow) off the Ebro Delta for domain A (350 m).**


The initial and boundary conditions for the coastal domains were obtained from two different CMEMS
products (IBI and MED). For the hydrodynamic module, hourly barotropic currents and sea levels are
consistently accommodated to the open boundaries with Chapman and Flather algorithms, whereas the
variability of currents along the water column (baroclinic component), temperature and salinity are imposed
from the CMEMS-IBI daily average values (or hourly data from CMEMS-MED) with clamped conditions.
The initial state of the smaller domains is obtained by interpolation from the larger domain conditions.

The comparison between the HFR and modelled eastward and northward components of the surface
currents (Figure A2) revealed good agreement and correlation between both datasets, both in intensity and
phase, and for both spatial components. The daily oscillations correspond to the inertial period in the region
(~19h) and are well reproduced by the model. Some intensifications of the currents -probably related with
energetic wind events- are also well described by the model. For comparison, Figure A2 also plots the
current components predicted by CMEMS-MED. For this particular period, the correlation between
measured and modelled data shows an $r^2 = 0.63$, slightly larger than the correlation between CMEMS-MED
data and measured values.

Regarding the currents within the bay, the fit between the modelled and measured values is shown in
Figure A3. Here, the general trend of the water flow is well captured by the model, which adequately
reproduces the main events, in spite of the very low energy of the system. This is a characteristic of both
Ebro Delta bays, Fangar and Alfacs (Cerralbo et al., 2014, 2019) although in Fangar it is enhanced by the
bay's shallowness and narrow connection with the open sea. For these verification numerical exercises, the
6-hourly ECMWF data has been used for the atmospheric forcing. For the hydro-biological simulations
shown in this contribution, the 70 m grid has been used because encompasses both the bay and part of the
outer area.

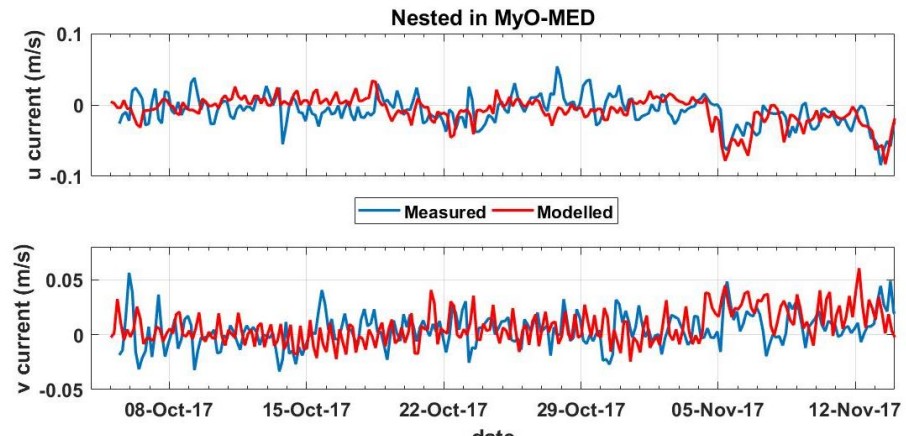

**Figure A3. Surface current components U (east-west) and V (north-south) measured inside the bay (blue) and modelled by the Fangar Bay nested scheme (red) during the 2017 autumn field campaign for domain C (23 m). Correspond to boundary forcing provided by CMEMS-MED.**

**7. Appendix B.** NPZD Model embedded in ROMS model**.**
The NPZD model follows a simple nitrogen-based scheme in order to simulate the interactions of the
four variables: nutrients (N), phytoplankton (P), zooplankton (Z) and detritus (D) (Figure B1). The
mathematical formulation of the internal fluxes varies in kind and complexity (see review Heinle & Slawig,
2013). This annex presents the equations used by the ROMS-NPZD model, as well as the values used for
the different parameters, based on Llebot et al. (2010), in the numerical simulations (Table B1).

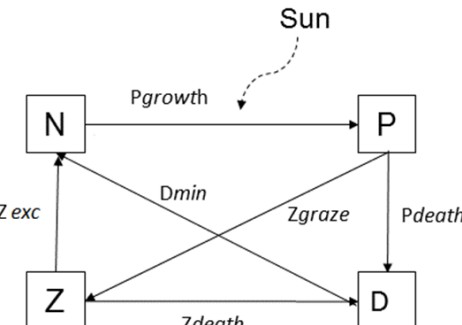


**Figure B1. ROMS-NPZD model scheme including the transfer functions of the different components. N**
**(nutrients), P (phytoplankton), Z (zooplankton) and D (detritus); Pdeath and Zdeath is phytoplankton and**
**zooplankton mortality respectively; Pgrowth is phytoplankton growth; Zexc is zooplankton exudation; Zgraze**
**is zooplankton grazing and Dmin is remineralization.**
**Nutrients**

$$\frac{\partial N}{\partial t} + \mathbf{u} \cdot \nabla N = \delta D + \gamma_n GZ - UP + \frac{\partial}{\partial z}\left(k_v \frac{\partial N}{\partial z}\right), \qquad \text{(B1)}$$

U = photosynthetic growth and uptake of nitrogen by phytoplankton
P = phototrophic phytoplankton
Z = herbivorous zooplankton





G = grazing on phytoplankton by zooplankton
$\gamma_n$ = some proportion of the consumed phytoplankton being lost directly to the nitrate pool as a function of
''sloppy feeding'' and metabolic processes.
**Phytoplankton**

$$\frac{\partial P}{\partial t} + \mathbf{u} \cdot \nabla P = UP - GZ - \sigma_d P + \frac{\partial}{\partial z}\left(k_v \frac{\partial P}{\partial z}\right), \quad \text{(B2)}$$

U = photosynthetic growth and uptake of nitrogen by phytoplankton
P = phototrophic phytoplankton
Z = herbivorous zooplankton
G = grazing on phytoplankton by zooplankton
$\sigma_d$ = phytoplankton mortality
**Zooplankton**

$$\frac{\partial Z}{\partial t} + \mathbf{u} \cdot \nabla Z = (1 - \gamma_n)GZ - \zeta_d Z + \frac{\partial}{\partial z}\left(k_v \frac{\partial Z}{\partial z}\right), \quad \text{(B3)}$$

Z = herbivorous zooplankton
G = grazing on phytoplankton by zooplankton
$\zeta_d$ = zooplankton mortality
$\gamma_n$ = some proportion of the consumed phytoplankton being lost directly to the nitrate pool as a function of
''sloppy feeding'' and metabolic processes.
**Detritus**

$$\frac{\partial D}{\partial t} + \mathbf{u} \cdot \nabla D = \sigma_d P + \zeta_d Z - \delta D + w_d \frac{\partial D}{\partial z} + \frac{\partial}{\partial z}\left(k_v \frac{\partial D}{\partial z}\right), \quad \text{(B4)}$$


**Grazing** $$G = R_m\left(1 - e^{-\Lambda P}\right), \quad \text{(B5)}$$

**Uptake** $$U = \frac{V_m N}{k_N + N}\frac{\alpha I}{\sqrt{V_m^2 + \alpha^2 I^2}}. \quad \text{(B6)}$$

**Irradiance** $$I = I_0 \exp\left(k_z z + k_p \int_0^z P(z')dz'\right), \quad \text{(B7)}$$

$K_z$ = light extinction coefficient
$k_p$ = self-shading coefficient





**Table B1. Parameters.**

| Parameter name | Symbol | Value | Dimension |
|---|---|---|---|
| Light extinction coefficient | $k_z$ | 0.067 | $m^{-1}$ |
| Self-shading coefficient | $k_p$ | 0.08 | $m^2$/ mmol-N |
| Initial slope of P-I curve | α | 0.025 | $m^2$ W$^{-1}$ |
| Surface irradiance | $I_o$ | 158.075 | W $m^{-2}$ |
| Nitrate uptake rate | $V_m$ | 1.5 | d$^{-1}$ |
| Phytoplankton mortality rate (senescence) | $\sigma_d$ | 0.15 | d$^{-1}$ |
| Uptake half saturation | $k_N$ | 0.8 | mmol-N $\cdot$ $m^{-3}$ |
| Zooplankton excretion efficiency | $\gamma_n$ | 0.03 | - |
| Zooplankton mortality rate | $\zeta_d$ | 0.08 | d$^{-1}$ |
| Ivlev constant | Λ | 0.06 | $m^3$ mmol·N$^{-1}$ |
| Zooplankton grazing rate | $R_m$ | 1.0 | d$^{-1}$ |
| Detritus remineralization rate | δ | 0.1 | d$^{-1}$ |
| Detrital sinking rate | $w_d$ | 5.0 | md$^{-1}$ |

**8. Acknowledgements**
The authors want to acknowledge the Ecosistema-BC Spanish research project (CTM2017-84275-
R/MICINN-AEI-FEDER, UE), ECO-BAYS research project (PID2020-115924RB-I00, financed by
MCIN/AEI/10.13039/501100011033) and the European Maritime and Fisheries Fund (EMFF) and the
Fisheries Directorate of the Catalan Government through the project ARP029/18/00008 Carrying capacity
for shellfish aquaculture in Fangar Bay. Also the CURAE project financed by Mercator Ocean International
(CMEMS Service Evolution 66-SE-CALL2), and thank Jordi Cateura and Joaquim Sospedra (LIM-UPC,
Barcelona, Spain) and the technical staff from IRTA for the data acquisition campaigns. As a group, we
would like to thank the Secretary for Universities and Research of the Department of Economy and
Knowledge of the Generalitat de Catalunya (2017SGR773).

**9. Author Contributions:** Conceptualization, methodology, M.F-P.B., M.G., M.E., M.M. and M.F-T.;
software, M.F-P.B., M.G., M.F-T., M.M. and M.E.; validation, M.F-P.B., M.M., M.G., M.E. and M.F-T.;
formal analysis, M.F-P.B.; investigation, M.F-P.B., M.G., M.E., M.M. and M.F-T.; resources, M.G., M.E,
M.M. and M.F-T.; data curation, M.F-P.B., M.M. and M.F.-T.; writing—original draft preparation, M.F-
P.B.; writing—review and editing, M.G., M.E., M.M., M.F.-T. and A.S-A; visualization, M.G.;
supervision, M.G. and M.E.; project administration, M.E., M.G., M.F-T. and A.S.-A; funding acquisition,
M.E., M.F-T and A.S-A. All authors have read and agreed to the published version of the manuscript.





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
