# Peer review of "Biological response to hydrodynamic factors in estuarine- # coastal systems: a numerical analysis in a micro-tidal bay."

_Biogeosciences, 2021_

## Referee Comment (RC1)

**Review: Biological response to hydrodynamic factors in estuarine-coastal systems: a numerical analysis in a micro-tidal bay.**

In this manuscript F-Pedrera Balsells and colleagues present the results of a numerical modelling exercise in which the role of the wind and land-based freshwater discharge on the spatial distribution of phytoplankton in a microtidal bay is explored (Fangar Bay, SP). This is done by altering starting conditions and forcings on an existing coupled ROMS-NPZD model of said bay. In short, the authors conclude that the wind direction and intensity do indeed play a role, potentially causing large heterogeneity in phytoplankton concentrations in what is a relatively small bay. The manuscript is very clearly written with equally clear figures. The aims of the study are well articulated and are also addressed in the results and discussion. Since the modelling setup has previously been published I assume that the model represents the system well. However three things are lacking for me: (1) more validation of the results coupled to (2) a longer reported output of the model, and (3) some sort of ecological discussion beyond a mention of the biomass.

1. The validation of experiments is presented to the reader as two figures of satellite output (figure 5). These images to me personally are not very convincing and I would like to see more satellite images or field data and model output be compared, possibly also through difference plots such as figure 4a. If a time series of photos can be found, then perhaps multiple successive satellite images can be compared to show how realistic the model output also is when a longer output is considered.

2. Now the output of 5 days is presented, but it should be longer. Looking at figure 2 D for example, the phytoplankton concentration in M4 (UW12) is increasing exponentially. When I see this, I wonder how high the phytoplankton concentration gets on such a short timespan and whether this is realistic. What are the peak concentrations that are reached in this bay in the year? The dynamics of zooplankton are only mentioned, since the output is too short to show the lagged response of zooplankton the phytoplankton availability. Also, once such a wind-event occurs and all the nitrogen has been depleted, how long does it take before a second bloom occurs? Or how long before the dynamics return to "normal" for example? These are all dynamics that can be explored in this manuscript to make it more fulfilling. Some of these things are mentioned in the conclusion as "future work", but since the modelling setup was all there from previous work I wonder why some of it was not already included (e.g. including P).

3. In the first sentences of the introduction a mention is made of socio-economic services, problems caused by aquaculture, ecosystem value,... Please couple back to this in the discussion. Because of the very short output of results, and a no discussion of the meaning in a biogeochemical sense, it is not clear why the results are important.

Please find minor remarks below.

L20: no dynamic

L35-36: This sentence seems out of place here.

L39: may control the inner water

L44: Please add older references here describing this phenomenon.

L48: create hypotheses and numerical experiments

L51: What is meant with "the biological mechanisms"?

L54-55: …an important source of both organic and inorganic nutrients for coastal areas.

L59: a larger variability compared to other *Mediterranean* coastal domains (?). In more northern latitudes such seasonality is highly common. Please indicate here also the phytoplankton values that can be expected throughout the year, and how they usually fluctuate.

L68: spatial-temporal -> spatio-temporal.

L77-82: This paragraph is not necessary as this is obvious from the text itself.

L87: I don't understand how the delta can reach 25 km offshore, please clarify this.

L108: How does nutrient input fluctuate with this opening and closing of the canals?

L129: A set of numerical experiments was…

L136: double brackets around the references.

L145: coupled with the ROMS model

L151: What is the mole fraction of Chl a? I do not see it in the table.

L155: Six experiments were designed with varying wind intensity and direction, and varying freshwater input from channels.

Remaining questions about the model after reading the materials and methods and appendix:

- Does it include wave activity?
- There is a finer scale 23 m grid available, why was this not used? Please also show the performance of the 70 m  grid in the appendix as validation.
- Do the channel inflows represent a nutrient input in ROMS as well?
- Are processes such as sediment input and water column turbidity included (referred to in line 270) and if so do they affect the light profiles used in the NPZD model?

L182: Four points within the bay were chosen

L187: consistently - > consistent

Figure 2: Why are the results of UW12fr not shown?

L195-196: Remove "for strong wind episodes".

L202: Leading to a larger presence

L204: lesser -> less

L208: perhaps change "comparison" to "similarity"? Also suggestions are best kept for the discussion.

L210: Please show longer output so that these differences in growth rates can be observed by the reader as well.

L239: Is it possible to add a panel to figure 4 with the nitrate concentrations? I wonder if the increasing phytoplankton concentrations may also be caused by nutrient inputs in the plumes.

L271: Again, please discuss how these factors are included in the ROMS-NPZD model and how they have affected the phytoplankton dynamics.

L274: is complex due to its intricate bathymetry. (its shallowness is part of the bathymetry already).

L275-281: please move this part to the introduction to better frame the study.

L281: What is meant with "the bio-hydrodynamics are well defined" ?

L282-283: Where is the magnitude of currents and the formation of fronts discussed in the results section?

L293: the presence of sea breezes -> the absence of sea breezes ?

L304: Non-uniformly -> non-uniform / heterogeneous.

L307: What is this classification, why does it matter, and in which typologies may Fangar bay be included?

L316-317: Which data is being interpreted?

L325: Ultimately, in a small, shallow,….

L329: So please show also plots of the nitrate concentrations as suggested previously. It is now not really clear whether the plume effect is only due to the nutrient inputs or also the salinity differences. It would be interesting to also discuss which phytoplankton species are present: are these marine species or more estuarine adapted species that thrive towards fresh water?

L331: Chl a concentrations are higher in… and decrease as salinity decreases.

L350: Which *in situ* observations are presented in the manuscript?

Figure 6: Very nice figure to explain what is going on. I suggest to use it also in the discussion to clarify processes.

L373: I am missing more evidence of these different phytoplankton patterns in need of an explanation, besides the two satellite pictures shown.

---

## Author Comment (AC1)

In this manuscript F-Pedrera Balsells and colleagues present the results of a numerical modelling exercise in which the role of the wind and land-based freshwater discharge on the spatial distribution of phytoplankton in a microtidal bay is explored (Fangar Bay, SP). This is done by altering starting conditions and forcings on an existing coupled ROMS-NPZD model of said bay. In short, the authors conclude that the wind direction and intensity do indeed play a role, potentially causing large heterogeneity in phytoplankton concentrations in what is a relatively small bay. The manuscript is very clearly written with equally clear figures. The aims of the study are well articulated and are also addressed in the results and discussion. Since the modelling setup has previously been published I assume that the model represents the system well. However, three things are lacking for me: (1) more validation of the results coupled to (2) a longer reported output of the model, and (3) some sort of ecological discussion beyond a mention of the biomass.

The authors acknowledge the helpful comments and corrections of Reviewer 1, which helped to improve the quality of the manuscript. Below, each comment is answered point-by-point. We have analyzed your comments carefully.  We have included the main corrections in the manuscript and the response to the reviewers' comments which we hope to meet with your approval. The answers or explanations are written in blue, and the newly added contents are in italics blue.

1. The validation of experiments is presented to the reader as two figures of satellite output (figure 5). These images to me personally are not very convincing and I would like to see more satellite images or field data and model output be compared, possibly also through difference plots such as figure 4a. If a time series of photos can be found, then perhaps multiple successive satellite images can be compared to show how realistic the model output also is when a longer output is considered.

We agree that the quality of these satellite images are not as we desired. The images shown in the paper are from a selection process of post-wind events. This means that the potential figures are somewhat limited and many of them do not present enough clarity because of cloud cover. Other images are shown in F-Pedrera Balsells et al. (2021), with similar "low quality". As you suggest the coupled model validation may be weak, so in the new version of the manuscript we include a qualitative validation of model outputs and *in situ* measurements which, jointly with remote sensing, we think provides credibility to our analysis. We also are considering improving these images for future work, using new satellite products combined with additional *in situ* field campaign oriented mainly for post-wind events.

2. Now the output of 5 days is presented, but it should be longer. Looking at figure 2 D for example, the phytoplankton concentration in M4 (UW12) is increasing exponentially. When I see this, I wonder how high the phytoplankton concentration gets on such a short timespan and whether this is realistic. What are the peak concentrations that are reached in this bay in the year? The dynamics of zooplankton are only mentioned, since the output is too short to show the lagged response of zooplankton the phytoplankton availability. Also, once such a wind-event occurs and all the nitrogen has been depleted, how long does it take before a second bloom occurs? Or how long before the dynamics return to "normal" for example? These are all dynamics that can be explored in this manuscript to make it more fulfilling. Some of these things are mentioned in the conclusion as "future work", but since the modelling setup was all there from previous work I wonder why some of it was not already included (e.g. including P).

The complexity of the hydrodynamic and biological variables suggests to face the analysis using "idealized" or "simplified" conditions instead of realistic (and long-term) simulations. We proved that these short simulations (of the order of the wind duration events) have been useful to understand the main hydro-biological coupled processes. It is true that more than 5 days would be required to simulate the evolution of the biological variables, but we wanted to adjust the simulations to the duration of the wind events observed in the region (Ràfols et al. 2017), to explore the response to this phenomenon (i.e. short-term response). This wind duration seems to be too short, but in order to understand the fundamental processes and the link of biological and hydrodynamic variables this duration was enough. In addition, we have observed that the spatial and temporal variability of Chl *a* is quite complex consistent with other coastal embayments (Demers et al., 1979; Díez-Minguito & de Swart, 2020; Masson & Peña, 2009; Mishra, S.; Mishra, 2012), so, using time limited simulations we tried to minimize the uncertainty and complexity isolating simple mechanisms. We think that is approach is quite new and may provide guidelines for the investigations of other regions focusing in specific events instead long-term simulations. In this sense the Introduction and Discussion has been modified to include this relevant point. As you suggested as additional simulations have been carried out paying attention to long-term evolution. Simulations have been carried out for up to 10 days (F-Pedrera Balsells et al., 2020). These simulations become pointless according to the results of primary production as we mentioned in the new version of the manuscript (see L301-310). However, we don't include these figures (see Figure R1) to not confuse the reader. Figure R1 shows how the Chl *a* concentration reaches values up to 500 mg·m$^{-3}$ at some of the control points. Probably, the reason for these senseless results is based on the short duration of the simulations and the fact that the model is not taking into account phytoplankton consumption, so the values shoot up. It also does not take into account the interaction of nutrients, phytoplankton and zooplankton with outside the bay (Open Sea), so larger concentrations accumulate within the bay. Also, we have tested long simulations (10 days) where the wind stops after five days. We observed that these does not provide any direct conclusions and requires further analysis.

[Figure]

*Figure R1. Time series of the Chl a at different points of the bay: (a) M1, (b) M2, (c) M3 and (d) M4. The different colours show the different simulations with in function of the wind. Solid lines*

*show the numerical results at the sea surface, dashed line shows numerical results at the sea bottom.*

Maximum reported concentration of Chl *a* measured in Fangar Bay has been 25 mg·m$^{-3}$ in September 1983 at 4 m depth (Delgado 1987) and 11 mg·m$^{-3}$ in October 2005 (Quijano-Scheggia et al., 2008). Also, vertical variability of Chl *a* has been measured within the Bay of the order of 5 mg·m$^{-3}$, suggesting the influence of the forcing mechanics and its interaction with primary production (L264-267).

P limitation has not been introduced in this work because the model used, ROMS-NPZD, does not include this mechanism. This is a model based on the N cycle only. We explain this in lines 335-339.

3. In the first sentences of the introduction a mention is made of socio-economic services, problems caused by aquaculture, ecosystem value, … Please couple back to this in the discussion. Because of the very short output of results, and a no discussion of the meaning in a biogeochemical sense, it is not clear why the results are important.

The new version of the manuscript includes a new paragraph in the Discussion analysing the importance and the potential interest of our investigation in terms of the bay activities management (e.g aquaculture, early warning systems, operational products, natural based solution, etc.) which pointed out the importance of the results.

L 372-386: Added in the manuscript: *"Hydro-ecological coupled models can be useful in the characterization of the evolution and prediction of nutrient variables as a tool of aquaculture management. Cerralbo et al. (2019) suggest the need to implement numerical tools in Ebro delta bays for early warning systems to prevent eventual mussel mortality during summer. Moreover, it is possible to combine this type of models, where the biogeochemistry of the bay is analyzed together with the hydrodynamics, with simpler models such as those of carrying capacity (Weitzman and Filgueira 2020; Guyondet et al., 2022) for better aquaculture management including harvest planning and early warning systems to avoid mortality (Hargreaves 1998; Yu and Gan 2021). They can even be extended to socio-economic study models of the area to cover all aspects related to aquaculture activity. Also, the use of hydrodynamic and biological models supports Nature Based Solutions (NBS) as an alternative to traditional engineering, with growing relevance to design integrated solutions for building coastal bay resilience (Pontee et al.,2016; F-Pedrera Balsells et al. 2020b) under climate change. Initial set of environmentally adapted alternatives in Fangar Bay may be: i) self-regulating connection with the open sea, ii) adjustable connection with land discharges or iii) adaptive reallocation of aquaculture activities; whose will require specific investigations on the hydro-biogeochemical response "*

**Minor remarks**.

L20: no dynamic

Corrected. Thanks.

L35-36: This sentence seems out of place here.

Corrected. Thanks.

L39: may control the inner water

Corrected. Thanks.

L42: Please add older references here describing this phenomenon.

Added in the text: de Madariaga, 1995.

L48: create hypotheses and numerical experiments

Corrected. Thanks.

L50: What is meant with "the biological mechanisms"?

Changed in the text to clarify: "physical mechanisms and biological behaviour mechanisms become more complex due to the geometry of the basin itself"

L54-55: …an important source of both organic and inorganic nutrients for coastal areas.

Corrected. Thanks.

L59: a larger variability compared to other Mediterranean coastal domains (?). In more northern latitudes such seasonality is highly common. Please indicate here also the phytoplankton values that can be expected throughout the year, and how they usually fluctuate.

Changed in the text to clarify: *In this sense, Chl* a *concentrations in Fangar Bay tend to show a distinct seasonal fluctuations entailing larger variability in comparison to other coastal domains such as the Ría de Arousa (Ramón et al., 2007) or Alfacs Bay (Artigas et al. 2014)* (L57-60).

L68: spatial-temporal -> spatio-temporal.

L70: Corrected. Thanks

L77-82: This paragraph is not necessary as this is obvious from the text itself.

OK, so we have removed it from the text.

L87: I don't understand how the delta can reach 25 km offshore, please clarify this.

The Delta itself, the deltaic sediments that form it, covers an area of 25 km (figure R2). However, if this information leads to confusion, we have decided to delete this sentence so as not to confuse the reader.

[Figure]

*Figure R2. Distance of the Delta offshore as seen from the satellite*

L108: How does nutrient input fluctuate with this opening and closing of the canals?

When the channels are closed (because drying process of the rice fields), there is no new input of nutrients from freshwater, so the bay functions only with the nutrients present in the water and those that can be supplied from the open seawater that enters from the bottom.

L129: A set of numerical experiments was…

L130: Corrected. Thanks

L136: double brackets around the references.

L138: Corrected. Thanks

L145: coupled with the ROMS model

L147: Corrected. Thanks

L151: What is the mole fraction of Chl a? I do not see it in the table.

L153: Right. Added in the text: 893.51 g/mol.

L157: Six experiments were designed with varying wind intensity and direction, and varying freshwater input from channels.

Remaining questions about the model after reading the materials and methods and appendix:

- Does it include wave activity?

No, we did not introduce wave activity into the model simulations. Here is a graph of the significant swell height in the Fangar Bay. The maximum swell we can find are waves of 0.5m (see Figure below). Moreover, to reach this order of magnitude, a very strong wind must be blowing, and the duration of this swell is only a few hours, so does not seems very relevant for phytoplankton dynamics.

[Figure]

*Figure R3. Significant wave heights in Fangar Bay*

- There is a finer scale 23 m grid available, why was this not used? Please also show

the performance of the 70 m grid in the appendix as validation.

In previous tests of the modelling scheme performance, it was noticed that the results of the 23 m grid did not provide significantly different results from those of the 70 m grid, although it did imply an important increase of the computational time. We have corrected the text to avoid confusion, removing the "23 m" paragraph.

- Do the channel inflows represent a nutrient input in ROMS as well?

Yes, both freshwater inputs from the drainage channels have a nutrient input in the model (L169-170).

- Are processes such as sediment input and water column turbidity included (referred

to in line 270) and if so do they affect the light profiles used in the NPZD model?

No, sediment input and turbidity are not taken into account in our simulations. Chlorophyll dynamics are quite complex in this area and the purpose is to simplify the processes in order to obtain robust conclusions. Also, the biochemical model considers simplified processes, which the equations are photosynthetic growth and nitrogen uptake by phytoplankton, grazing of phytoplankton by zooplankton, mortality of phytoplankton and zooplankton, and sinking and remineralisation of detritus. Suspended sediment and turbidity entails wave and sediment transport mechanics which are included in the future work (see L417-L421).

L182: Four points within the bay were chosen

L184: Corrected. Thanks.

L187: consistently - > consistent

L189: Corrected. Thanks.

Figure 2: Why are the results of UW12fr not shown?

We considered that for what we want to show in this figure (wind event response), the results of the UW12fr (freshwater discharge) simulation were not relevant.

L195-196: Remove "for strong wind episodes".

Corrected. Thanks.

L202: Leading to a larger presence

Corrected. Thanks.

L204: lesser -> less

Corrected. Thanks.

L208: perhaps change "comparison" to "similarity"? Also suggestions are best kept for the discussion.

Corrected. Thanks.

L210: Please show longer output so that these differences in growth rates can be observed by the reader as well.

L202: As mentioned above, the approach of this study has been to simplify the processes using short simulations, due to the complexity of the chlorophyll evolution and the hydrodynamics of the area. Thanks to our previous work (F-Pedrera Balsells et al., 2020, 2021) we have been able to quantify the intuitive behaviour of the dynamics within the bay, so our manuscript pursues to take advantage of this skills to face simplified numerical modelling in order to investigate the Chl $a$ as a function of the different winds. We think that is approach is quite new and may provide guidelines for the investigations of other regions.

L239: Is it possible to add a panel to figure 4 with the nitrate concentrations? I wonder if the increasing phytoplankton concentrations may also be caused by nutrient inputs in the plumes.

Here we show the $NO_3$ concentration. As it can be seen in Figure R3, there is nutrient input at the initial moment (continuous lines) and then there is a consumption by phytoplankton that reduces this concentration (dashed lines). With this simple model, where there is no further contribution of nutrients neither by the suspended sediment nor by the input from the open sea they are almost depleted by the end of the simulation. Added in the manuscript, L246-250.

[Figure]

*Figure R4. Nitrates concentration. The solid lines show the initial concentrations and the dashed lines show the final concentrations*

L271: Again, please discuss how these factors are included in the ROMS-NPZD model and how they have affected the phytoplankton dynamics.

As mentioned above, the NPZD model only accounts for nitrate as a nutrient in addition to photosynthetic growth and nitrogen uptake by phytoplankton, grazing of phytoplankton by zooplankton, mortality of phytoplankton and zooplankton, and sinking and remineralisation of detritus. We suspect that the sediment resuspension and turbidity has influence on the dynamics but this would increase the complexity adding a sediment transport model. However, in discussion we draw some ideas to face this point in future works based on our current skills (see L414-419).

L274: is complex due to its intricate bathymetry. (its shallowness is part of the bathymetry already).

Corrected, thanks.

L275-281: please move this part to the introduction to better frame the study.

Done.

L281: What is meant with "the bio-hydrodynamics are well defined" ?

L79: In the previously published article (F-Pedrera Balsells et al, 2021) we explained the behaviour of surface chlorophyll concentration according to wind-driven currents.

L282-283: Where is the magnitude of currents and the formation of fronts discussed in the results section?

The influence of wind on currents can be seen in the freshwater plume shown in Figure 3b. The magnitude is also included in the new version of the manuscript (L297).

L293: the presence of sea breezes -> the absence of sea breezes?

This means that during periods of calm wind or even sea breeze winds, the phytoplankton biomass is governed by the estuarine circulation.

L304: Non-uniformly -> non-uniform / heterogeneous.

Corrected, thanks.

L307: What is this classification, why does it matter, and in which typologies may Fangar bay

be included?

L331-334: Jiang et al. (2020) in their paper entitled *'Drivers of the spatial phytoplankton gradient in estuarine-coastal systems: generic implications of a case study in a Dutch tidal bay'* classify estuaries according to the distribution of phytoplankton biomass. We refer to this article because it is one of those which classifies estuaries according to a biological parameter. With our results we have determined that the Fangar Bay can be classified according to this method in an estuary with an irregular biomass pattern, since depending on the wind blowing we obtain one pattern or another of distribution.

| Common spatial gradients | Example ecosystems and references | Flushing mechanisms | Main drivers of phytoplankton biomass |
|---|---|---|---|
| Seaward increasing | (1) Oosterschelde, the Netherlands (this study) | Tide dominated | Grazing loss and tidal import |
| | (2) Rías Baixas of Galicia, Spain (Figueiras et al., 2002); Willapa Bay, USA (Hickey and Banas, 2003; Banas et al., 2007) | Tide dominated | Tidal import |
| | (3) Chilika Lagoon, India (Srichandan et al., 2015) | River dominated | Light limitation |
| Seaward decreasing | (1) Westerschelde Estuary, the Netherlands and Belgium (Kromkamp and Peene, 1995; Krompkamp et al., 1995; Muylaert et al., 2005; Soetaert et al., 1994, 2006) | River and tides, or river dominated | Salinity stress, grazing loss, and transport |
| | (2) Chesapeake Bay outflow plume, USA (Jiang and Xia, 2018); Mississippi River plume, USA (Gomez et al., 2018) | River and tides | Nutrient limitation |
| Chl-a maximum | (1) Chesapeake Bay, USA (Jiang and Xia, 2017); Delaware Bay, USA (Fisher et al., 1988); York River, USA (Sin et al., 1999); Neuse-Pamlico Estuary, USA (Valdes-Weaver et al., 2006); Logan River and Moreton Bay, Australia (O'Donohue and Dennison, 1997) | River and tides, or river dominated | Upper reach limited by light or transport loss; lower reach limited by nutrients |
| Spatially uniform or weak spatial gradient | (1) San Francisco Bay, USA, after 1987 (Cloern et al., 2017; Kimmerer and Thompson, 2014) | River and tides | Grazing loss |
| | (2) Hudson River estuary, USA (Fisher et al., 1988; Howarth et al., 2000; Strayer et al., 2008) | River dominated | Transport and grazing loss |
| Patches/irregular patterns | (1) Baie des Veys Estuary, France (Grangeré et al., 2010) | River and tides | Grazing loss |
| | (2) Krka Estuary, Croatia (Ahel et al., 1996) | River dominated | Point-source nutrient input |
| | (3) St. Lucia Estuary, South Africa (van der Molen and Perissinotto, 2011) | River dominated | DIN:DIP ratio, salinity, temperature, and irradiance |

**Figure 13.** Common spatial patterns of phytoplankton biomass in estuarine–coastal systems. For comparison with the Eastern Scheldt, example ecosystems for each type are given along with references, the dominant flushing mechanisms, and main drivers of phytoplankton accumulation.

L316-317: Which data is being interpreted?

L337: An interpretation of the nitrogen data, as we do not take into account the phosphorus limitation in these simulations.

L325: Ultimately, in a small, shallow,....

Corrected, thanks.

L329: So please show also plots of the nitrate concentrations as suggested previously. It is

now not really clear whether the plume effect is only due to the nutrient inputs or also the salinity differences. It would be interesting to also discuss which phytoplankton species are present: are these marine species or more estuarine adapted species that thrive towards fresh water?

Added in figure 4 in the manuscript.

L331: Chl a concentrations are higher in… and decrease as salinity decreases.

Corrected, thanks.

L350: Which in situ observations are presented in the manuscript?

L385: Right, in this manuscript, there are no *in situ* observations. Thanks.

Figure 6: Very nice figure to explain what is going on. I suggest to use it also in the discussion to clarify processes.

We agree. We have moved the figure 6 to the Discussion and added a sentence (L328).

L373: I am missing more evidence of these different phytoplankton patterns in need of an explanation, besides the two satellite pictures shown.

L411. Below are shown all the available Sentinel 2 images for the study period. Here is a summary of some of them where you can see the same behaviour in the two situations we mentioned: downwind and upwind periods. We did not add more figures to not confuse the reader.

**Sea breeze (downwind)**

[Figure]

**NW winds (upwind)**

---

## Author Comment (AC2)

**General comments**

This paper deals with the influence of wind and freshwater discharge over the dynamics and phytoplankton distribution in a microtidal bay (Fangar Bay). Modeling experiments using a coupled physical-biogeochemical model with varying wind and freshwater input are used to show the large variability in phytoplankton concentration in the bay depending on the wind intensity and direction.

This paper is well written and clear. Figures are of good quality and a conceptual diagram of the processes affecting the Fangar bay is proposed.

The authors acknowledge the helpful comments and corrections of Reviewer 2, which helped to improve the quality of the manuscript. Below, each comment is answered point-by-point. We have analyzed your comments carefully. We have included the main corrections in the manuscript and the response to the reviewers' comments which we hope to meet with your approval. The answers or explanations are written in blue, and the newly added contents are in italics blue.

The authors previously published a few papers investigating the hydrodynamics and biogeochemistry in the Fangar Bay, and this paper is presented as a complement to this previous series. In the 2020a paper, the authors used in situ and model data to study the hydrodynamics of the bay and to assess the influence of bathymetry on wind-driven circulation, the wind being constant. In the 2020b paper, the influence of river outflow on residence time is assessed. In the 2021 paper, in situ data of hydrodynamics and Chlorophyll-a are used to show the influence of the wind and stratification on Chl-a surface pattern. However, even if there is an attempt of explanation, more should be made on how this paper brings substantial new results compared with the papers 2020a, 2020b and 2021. Results on phytoplankton biomass could also be emphasized for their meaning in explaining the ecosystem functioning of the bay.

We agree that the improvement and/or specific contribution of this paper is not sufficiently highlighted in the current version. The third paragraph of the introduction has been modified. The objective has been clarified and the link with previous works (e.g. F-Pedrera Balsells et al., 2021 and Llebot et al., 2011) rewritten. So, the objective of this work is now explained in lines 75-77.

The meaning and application of the ecosystem function in the bay is also explained in the new version of the manuscript (see new paragraph in L77-82). It is possible to combine this type of models, where the biogeochemistry of the bay is analysed together with the hydrodynamics, with simpler models such as those of carrying capacity (Weitzman and Filgueira 2020; Guyondet et al., 2022) for better aquaculture management including harvest planning and early warning systems to avoid mortality (Hargreaves 1998; Yu and Gan 2021). They can even be extended to socio-economic study models of the area to cover all aspects related to aquaculture activity. Also, the use of hydrodynamic and biological models supports Nature Based Solutions (NBS) as an alternative to traditional engineering, with growing relevance to design integrated solutions for building coastal bay resilience (Pontee et al.,2016; F-Pedrera Balsells et al. 2020b) under climate change. Initial set of environmentally adapted alternatives in Fangar Bay may be: i) self-regulating connection with the open sea, ii) adjustable connection with land discharges or iii) adaptive reallocation of aquaculture activities; whose will require specific investigations on the hydro-biogeochemical response (L372-386).

Another important concern is the model validation. Why showing only model/data comparisons at the surface (with satellite imagery for phytoplankton concentration) although the bay can be stratified, the surface/bottom biogeochemistry is contrasted and the model results are also explored at the bottom? In the previous papers 2020a and 2021, in situ data time series are available in summer and autumn along the water column, for physical and biogeochemical variables. Some of the wind situations shown in the present paper may have been encountered during the in situ experiments (indeed this is the case for NW winds).

About the vertical variability, it has been observed in other works (Ramón et al., 2007; Soriano-González et al., 2019) that when strong winds blow, the surface data are similar to the integrated data of the whole water column consistent with our model results. In addition, it should be noted that this is a very shallow bay (4m max) and it is often difficult to distinguish the surface/bottom layers in the sampling. Our work focuses on spatial and temporal distribution instead of vertical variability which also is of great importance and not still understood in very shallow domains.

The new version of the manuscript includes a new qualitative comparison with numerical results and observations including order of magnitudes of the vertical variability (L256-267) using specific conversion factors. Throughout the discussion, previous work is cited and compared with the new results provided in this manuscript. E.g. lines 287-292, 294-298, 328-330.

How does the model handle realistic situations close to the idealized experiments in this paper, why not considering realistic cases for comparison with in situ data? Is this left to a future paper? I suggest to add a discussion on how the results of the previous paper compare with previous observations.

For this investigation of the influence of the physical variables on the biological variables, we decided to show "idealized" or "simplified" simulations results where the wind was constant but varying the direction in order to observe the dynamics of phytoplankton biomass as a function of wind throughout the water column. This is because the system is extremely complex in terms of hydrodynamic and biological variables as we stated in previous papers and similar domains (i.e. coastal shallow embayment's and small estuaries) leading strong temporal and spatial variability (second paragraph of the Introduction reflects this behaviour and also encountered in the cited works: Demers et al., 1979; Díez-Minguito & de Swart, 2020; Masson & Peña, 2009; Mishra, 2012). To run "idealized" (i.e., controlled) conditions has proved more effective to understand the processes in contrast to using realistic cases (see F-Pedrera Balsells et al., 2020a).

However, we agree that this is a critical point that must be clarified in the new version of the manuscript. We have added two additional sentences highlighting this point in the Discussion which we think that may help the reader to understand the strategy used in our investigation (L298-309). Additional simulations have been carried out paying attention to long-term evolution (10 days). These simulations show pointless results (see figure R1), and even though we mention the analysis in the new version of the manuscript (see L301-310) we preferred not to extend this point to not confuse the reader. Figure R1 shows that the Chl $a$ concentration reaches values up to 500 mg·m$^{-3}$ at some of the control points. Probably, the reason for the senseless results is based on the fact that the model is not taking into account the phytoplankton consumption, so the values increase. It also does not take into account the interaction of nutrients, phytoplankton and zooplankton from inside the bay to the open sea. Also, we have tested long simulations (10 days) where the wind stops after 5 days. We have observed that it does not provide any direct conclusions and requires further analysis.

[Figure]

*Figure R1. Time series of the Chl a at different points of the bay: (a) M1, (b) M2, (c) M3 and (d) M4. The different colours show the different simulations with in function of the wind. Solid lines show the numerical results at the sea surface, dashed line shows numerical results at the sea bottom.*

Second paragraph of the Conclusions also addresses this point of using idealized simulations for an accurate analysis. As future work a new sentence has been added including the realistic simulations configurations set-up (see L423-426).

I would recommend publication in Biogeosciences only if the previous and following comments are adequately addressed.

**Specific comments**

L. 136-137: In the introduction, the authors say that the model has been validated in the Fangar Bay, and refer to the appendix. This may mislead the reader, as the validation is presented only in this paper and was not done before.

As we pointed out before, the analysis is supported on idealized runs to discern the driving mechanics of the hydro and biological evolution. Because we are not using realistic (long-term) simulations results we think that including the validation in the core of the paper may mislead the reader. We think that the previous comments about idealized vs realistic simulations may also help the reader.

L. 72: is there really a biological model? I think that you mean biogeochemical model. You claim that it is embedded into the hydrodynamical model, is it not rather coupled or forced?

L77: Ok, changed. It is a coupled model. Seems more suitable than forced, because the NPZD scripts run online is not "externally" forced.

L. 101: the authors mention NE winds of great intensity in the bay. However, on page 4 (L.159) and in Figure 6, the wind is SE

That is a typing error. The correct wind is SE. Corrected already in the text.

Figure 1: Please add the position of control points M1-M4 in a separate Table.

Added in the manuscript.

|  | Latitude (º) | Longitude (º) | Depth (in m) |
|---|---|---|---|
| **M1** | 40.775306 | 0.720305 | 4.05 |
| **M2** | 40.767762 | 0.742785 | 4.02 |
| **M3** | 40.771534 | 0.735841 | 1.79 |
| **M4** | 40.758125 | 0.771917 | 0.93 |
| **BO** | 40.785970 | 0.709483 | - |
| **IM** | 40.766413 | 0.738546 | - |

L. 130-136: These lines are exactly the same than in 2020a (section 2.3, first paragraph). Please avoid copying the exact sentences from a previous paper.

Modified.

L. 133: How are the 10 sigma levels distributed on the vertical? How is the bathymetry set up, as in the previous paper 2020b you used an idealized bathymetry "due to the difficulty of achieving good bathymetry"?

L140: The 10 sigma levels are distributed according to the plot referenced in the article F-Pedrera Balsells et al. (2020b). Subsequent to that work we were able to obtain a more realistic bathymetry which is the one used for this work.

L. 151: What is the mole fraction of Chl-a?

L153: 893.51 g/mol. Added in the text.

L. 155 and Table 1: For experiment UW12fr, why choosing a channel flow of 3 m3/s?

This was a mistake: 3.75 is the proper value that is half the flow rate that usually flows out of the drainage channels (7.5 $m^3 \cdot s^{-1}$). Clarified also in the new version of the manuscript (L162).

The wind blows for 3 to 5 days, which explains the choice of the experiments. However, from Figure 2, the results can be quite different after 3 or 5 days for phytoplankton biomass. Tis point isn't discussed in the manuscript, I suggest that you add the impact of wind duration variability in your discussion.

The complexity of the hydrodynamic and biological variables suggests to face the analysis using "idealized" conditions instead of realistic (and long-term) simulations. Also, we proved that these short simulations (of the order of the wind duration events) have been useful to understand the main coupled hydro-biological processes. A simulation length of 5 days has been chosen because it corresponds to the average duration of the wind events (Ràfols et al. 2017). This wind duration seems to be too short, but in order to understand the fundamental processes and the link of biological and hydrodynamic variables this duration was enough. Results of larger simulations have been also analysed (10 days), but those become unfeasible according to the observations of primary production (see figure R1) (L301-310).

L. 161 and Figure 3: How is the initial stratification set up? You mention a previous paper (2021) as an explanation for the choice of your stratification profile. However it is not clear to me how you chose it. Is the salinity an average of the two profiles shown on Figure 6 of 2021 paper? I did

not find any T plots on this paper, except at the bottom. Please provide more information, and mention that the initial salinity profile is sown on figure 3 of the present paper.

L164: The ROMS model requires initial and boundary conditions data for both temperature and salinity. The initial and boundary values introduced in the simulations have been taken from the campaigns of the mentioned article (F-Pedrera Balsells et al., 2021), where you can see the bottom temperature values in figure 2 and 3 and the salinity values in figure 6, as you indicate. These values are not an average but the exact values, which have been interpolated in the numerical mesh.

Added in the manuscript, line 166-167 "*Figure 3 shows the horizontal distribution of modelled salinity based on initial conditions interpolated from the observation shown in F-Pedrera Balsells et al. (2021).*".

For freshwater fluxes: the authors take a constant value of 7.5 m3/s for most of the cases, without differentiating between cases of NW and SE winds. However, as stated by the authors, SE winds are associated with local rain events, could this induce an increase of the channels outflow, or are the fluxes only driven by the rice cultivation activities?

The freshwater flows in these channels are in principle only driven by rice cultivation activities. This chosen value is the average of the data measured in the channels (i.e. *in situ*). Likely heavy rainfall increases the freshwater flow and may generate an interesting situation within the bay (including eventual mixing and saltwater flushing), but unfortunately no observations are available during these events. In the new version of the manuscript this point has been included as future work (see second paragraph of the Discussion).

L. 183: add reference to "(Figure 1)" at the end of the sentence.

L188: Done. Thanks.

L. 188: the authors say that "all simulations show larger concentrations of phytoplankton biomass at the surface due to freshwater fluxes", and refer to later discussion. But at L.196-197, they say that "for the UW10 and UW12 simulations (moderate and strong upbay wind), both surface and bottom phytoplankton time series coincide at all control points ". Which of these two statements is correct?

L192-196: Right, this leads to confusion. The text has been amended to clarify this issue: "*stratification conditions (CALM and DW6 simulations) show higher phytoplankton biomass concentrations at the surface due to freshwater fluxes. Substantial differences in phytoplankton biomass between the surface and bottom layers are evident in M1, where stratification tends to be greater in contrast to the shallowest point (M4)*".

L. 210: The authors say that there is a difference in growth rates observed between phytoplankton and zooplankton, without any reference to a figure. Not shown? It could help to add a Figure for zooplankton (in Appendix?).

In this paper we focus in the primary production instead of zooplankton which has a larger growth rate in comparison to phytoplankton. The graphs do not show much of the data (Figure R2), so we have not included them in the manuscript.

[Figure]

*Figure R2. Zooplankton evolution at different control points for all the simulations.*

L. 241-242: I think that you refer to "Figure 4" instead of "Figure 3", and the difference shown by the figure may not be "UW10" at L. 242 because there is not any plot showing UW10 results, I guess it is "UW12". In this case, you should not write "winds of similar intensities" at L. 241.

The explanation refers to figure 4 (before figure 3). The vertical profiles of phytoplankton biomass show how winds of the same intensity (12 m·s$^{-1}$) but different direction (NW (UW12) - SE (DW8)) cause a completely different horizontal distribution of phytoplankton concentration. We have corrected the name of the simulation (UW12) in the text (L226).

Figure 4 gives the spatial pattern of phytoplankton biomass and salinity at the surface and bottom of the bay. For the reader, it is easier to understand the spatial gradients and the bottom/surface differences from this figure than from vertical profiles at control points. I suggest to place this figure before Figure 3. Indeed you first use Figure 3 to explore horizontal differences between control points and this should be placed after the general horizontal maps. Also, it is not clear to me why nitrates do not appear anymore.

Ok, changed. Nitrates at the end of the simulations also have been added in Figure 4. We modified also the manuscript according to these results: *As it can be seen in Figure 4c, there is nutrient input at the initial moment (continuous lines) and then there is a consumption by phytoplankton that reduces this concentration (dashed lines). With this simple model, where there is no further contribution of nutrients neither by the suspended sediment nor by the input from the open sea they are almost depleted by the end of the simulation* (L246-250)

L. 264: add "and nutrient" after "freshwater".

284: Done, thanks.

L. 274-281: This part of the discussion could be moved to the introduction to better explain the purpose of the present study.

Done, thanks.

L. 298: The difference in nutrients availability along the water column depending on the wind direction is not discussed from figure 2.

L321-323: *"When strong NW up-bay winds blow in Fangar Bay, the water column homogenises, making nutrients available throughout the column, both at the surface and at the bottom (i.e., UPW10 and UW12 simulations)".* → this can be seen by looking at the continuous and dashed lines, pink and green, overlapping at points M1, M2 and M4 in figure2.

L325-328: *"With strong SE down-bay winds (DW8 simulation), phytoplankton biomass increases near the discharge channels and the phytoplankton biomass distribution follows the water circulation driven by SE winds"* → The same is true for the DW8 simulation (brown line). The solid and dashed lines overlap showing how the water column homogenises when these winds blow.

L. 390-391: You mention the comparisons of modelled currents with observed current profiles in 2017. Where are the results for the comparison of currents along the water column? You should definitely add some model/observed data (shown in the 2020a paper for example) comparisons in the water column and not only at the surface.

Qualitative comparison of the idealised coupled simulations and *in situ* data and remote sensing provides robustness to our analysis. The increase of phytoplankton in the first days of the wind event numerical simulations are consistent with the conclusions drawn by F-Pedrera Balsells et al. (2021) which pointed out that wind episodes causes an increase in the concentration of surface Chl *a*. Chlorophyll *a* field data collection were obtained using seawater samples (F-Pedrera Balsells et al. 2021) and the range obtained after wind events were 4 mg·m$^{-3}$ and 7 mg·m$^{-3}$. These values agree with the values obtained after 5 days of wind simulations (e.g.~ 3 mmol·m$^{-3}$) assuming a 1.59 g chlorophyll per mole N conversion suggested by Gong et al. (2015) (L256-263). We can see in F-Pedrera Balsells et al. (2020b) that the magnitude of the current of the simplified model (figure 11) is similar to that of the observations (figure 6) (0.1 m·s$^{-1}$, added in L299), so we were able to make a qualitative assessment.

L. 389-393 and Figure A2: What is the spatial coverage of the HRF radar? What are U and V on the plot, is it a spatial average or at a particular location?

L 435-438: U and V are the zonal and meridional components of the current at a random offshore position. This has been specified now in the manuscript, and a reference to a paper describing the HFR system and its characteristics has also been added.

L. 402: is the initial states of smaller domains only obtained from interpolation or do also you have to perform extrapolation? If the answer is yes, please add it in the text.

L443: No. The initial state of the different domains is obtained by interpolation from the respective parent domains.

L. 413-414: I agree that the model reproduces well some events (in November especially), but the comparison is not so good for the U component around 15 October for example from Figure A2 and Figure A3. Moreover, Figure A3 shows that the v component is out-of-phase, and even if values are relatively small, the authors cannot validate the model from this figure. To compare the general trends, I would suggest to show filtered data sets. Also, adding statistics (RMS, mean, bias, correlation…) would be very useful to show the model validation (this can be added in a table).

L455-457: Yes, the authors agree that it is difficult to validate a model when both measured and simulated values are so small, and possibly within the error range of the instrument and the model for a significant portion of the observational period. In addition, measurement errors can

be aggravated due to the likely presence of very fine suspended sediment in the water column (Fangar loosely translates as "place of mud"). Under these conditions, small variations in the velocities can lead to large errors in model performance. We believe that a Lagrangian validation scheme would have been preferable, instead of a Eulerian approach, but this was not possible to do.

L. 418: the authors show a model validation for the 350m model resolution, is the comparison is for the Fangar Bay? (in this case, why not showing a comparison of HRF radar data with the embedded configuration B that is used in the present study (L. 418)?) This does not really make sense for me, the configuration used for the study is not validated in the paper. And there is no validation of hydrology, is there any SST image that could be used? There are hydrological data from the in situ campaigns.

459: The idea behind the validation procedure was to validate each one of the domains to ensure the robustness of the nesting scheme. There are no field data that can be used simultaneously for both grids, and there is some difficulty in knowing all the natural data needed to enter them into the model. On the one hand, the HFR spatial coverage does not include the bay (as can be seen in Lorente et al., 2015. *Evaluating the surface circulation in the Ebro delta (northeastern Spain) with quality-controlled high-frequency radar measurements*, Oc. Sci., 11(6), 921-935) and, on the other one, the 350-m grid is too coarse for us to expect a decent hydrodynamic description inside the bay. Therefore, a separate process validation has been carried out for domains A and B.

**Technical corrections**

Typos and spelling: All done, thanks for the corrections.

L.68: replace "spatial-temporal" by "spatio-temporal"

L. 65: replace "on" by "in"

L. 78: add "of" before "satellite images"

L.79: add "of" before "phytoplankton"

L. 137: replace "Annex 1" by "Appendix A"

L. 156: add "input" after "freshwater"

L. 215: replace "B1" by "Figure B1"

L. 229: add "input" after "freshwater"

L. 298: add "water" before "column"

Figures: All done, thanks for the corrections.

Figure 2: in the legend, replace "Solid lines shows surface numerical results, dashed line shows bottom numerical results" by "Solid lines shows the numerical results at the sea surface, dashed line shows numerical results at the sea bottom". It is very difficult to distinguish between the different curves on the plot. For a to c, bottom dotted lines are hardly distinguishable from surface curves. For all plot, the orange/red lines are hardly distinguishable. Please find a way to

improve the figure. The concentration in nitrates has a maximum value of around 3 mmol/m3, so I suggest that you take 3 or 4 mmol/m3 as a maximum in the plot.

Figure 3: Replace "vertical Chl a profiles" by "phytoplankton biomass" in the legend to be consistent with the text describing the results

Figure A2: add "horizontal resolution grid" after "350m".

---

## Author Comment (AC3)

[revised manuscript text omitted]
 and 79 provide answers to unresolved questions suggested by F-Pedrera Balsells et al. (2021) and Llebot et al. 80 (2010) in terms of spatial and temporal variability on Chl a in shallow and coastal bays. Extensive field 81 data and previous hydrodynamic knowledge converts Fangar Bay in a unique study area to investigate the 82 biological response in an area with large spatio-temporal variability in Chl a evolution.

**84 2. Material and Methods**

2.1. Study area

Fangar Bay is part of the Ebro Delta (NW Mediterranean Sea), which forms two semi-enclosed bays, Fangar to the north and Alfacs to the south. Of these, Fangar Bay is the smallest, extending over 12 km2, with a length of about 6 km, a maximum width of 2 km and a volume of water of 16·106 m3 (Delgado and Camp 1987). The average depth is 2 m, with a maximum of 4 m (see bathymetry in Figure 1). Its connection with the open sea is oriented to the NW, and is approximately 1 km wide (Garcia and Ballester 1984), although it is currently narrowing because of the accumulation of sediment from the beach located to the north (Archetti, Bernia, and Salvà-Catarineu 2010).

The wind regime in the Fangar Bay area is characterized by the presence of S/SE sea breezes – which 96 do not exceed 6 m·s-1 during spring and summer- and strong winds from the N and NW of more than 12 97 m·s-1 in autumn and winter (Bolaños et al. 2009; Grifoll et al. 2016). The most frequent wind throughout 98 the year is locally known as Mestral, which is characterized by strong gusts of cold and dry wind from the 99 NW (Garcia and Ballester 1984). These winds are associated to the general weather pattern and occur 100 throughout the year, but show maximal strength and persistence during the colder months. Additionally, E 101 and SE winds that can also be quite intense ( $\sim 10 \text{ m} \cdot \text{s}^{-1}$ ) are responsible for local rain events and transient 102 increases of the local mean sea level at the coast (Muñoz 1990).

Both the Ebro Delta bays receive freshwater inputs from the channels irrigating the Delta paddies. This 105 freshwater outflow is regulated by the rice cultivation cycle throughout the year. In Fangar Bay, the 106 channels are open between April and November, discharging a mean flow of 7.23 m3·s-1 (SAICA Project, 107 2013. Available online: https://www.saica.co.za/ (accessed on 30 January 2020)), whereas the outflow is 108 negligible from December to March, when the channels are closed (Perez & Camp, 1986). There are two 109 main freshwater discharges in Fangar Bay: one in the Illa de Mar harbour inside the bay (IM in Figure 1) 110 and the other one, Bassa de les Olles, located at the bay mouth (BO in Figure 1). In addition to these, 111 freshwater inputs are also expected inside the bay from groundwater sources (Camp and Delgado 1987), 112 and along the coastline where freshwater inflows regulated by gravity according to the sea level occur. In 113 both cases, the expected freshwater inflow is smaller than that discharged from the regulated irrigation 114 channels.

- 115
- 116

Figure 1. Location of the study area. The red circles show the two main points of freshwater discharges (Bassa de les Olles (BO) and Illa de Mar (IM). The yellow stars show the location of the control points used for the numerical model results (Table 1). The bathymetry is also shown in the figure.

Fangar Bay is micro tidal, with a tidal range smaller than 1 m, which accentuates the action of the wind, 119 and is stratified most of the year mainly due to the freshwater flows rather than to the atmospheric heat 120 fluxes. Because of its bathymetry and complex geometry there is a strong transverse variability of the water 121 flows, particularly for prevalent up-bay wind episodes (NW winds), during which up-bay flow occurs in 122 the lateral shoals and down-bay flow in the central channel for up-bay wind pulses. These winds also cause 123 homogenisation of the whole water column. On the other hand, during calm periods the water circulation 124 is complex: current velocities are very small and lack a clear pattern, and the bay is strongly stratified due 125 to the freshwater inputs from the drainage channels (F-Pedrera Balsells et al., 2020a).

|    | Latitude (º) | Longitude (º) | Depth (in m) |
|----|--------------|---------------|--------------|
| M1 | 40.775306    | 0.720305      | 4.05         |
| M2 | 40.767762    | 0.742785      | 4.02         |
| М3 | 40.771534    | 0.735841      | 1.79         |

| M4 | 40.758125 | 0.771917 | 0.93 |
|----|-----------|----------|------|
| BO | 40.785970 | 0.709483 | -    |
| IM | 40.766413 | 0.738546 | -    |

**129 2.2. Numerical model and experiments design**

To analyse the relationship between the hydrodynamic and Chl a response to wind in small and 131 shallow estuaries, the Regional Ocean Modelling System (ROMS) was used to perform a series of 132 numerical experiments. The ROMS numerical model is a 3D, free-surface, terrain-following numerical 133 model that solves the Reynolds-Averaged Navier-Stokes equations using hydrostatic and Boussinesq 134 assumptions (Shchepetkin and McWilliams 2005). To discretize the horizontal grid into curvilinear 135 orthogonal coordinates and finite difference approximations on stretched vertical coordinates, ROMS uses 136 the Arakawa-C differentiation scheme (Haidvogel et al. 2007). The numerical details of ROMS are described extensively in (Shchepetkin and McWilliams 2005). This model has been used and validated in 137 138 similar bays and estuaries, such as Alfacs Bay located south of the Ebro Delta (e.g. Cerralbo et al., 2014, 139 2015, 2019) and in the Fangar Bay (see Appendix 1). The domain used for the experiments consists of a 140 regular 107x147 grid with a horizontal resolution of about 70 m and 10 sigma levels in the vertical direction 141 (F-Pedrera Balsells et al. 2020a). The model boundary is located 10 nodes away from the bay's entrance to 142 avoid boundary noise. The hydrodynamic bottom boundary layer was parametrised with a logarithmic 143 profile using a characteristic bottom roughness height of 0.2 m. The turbulence closure scheme for the 144 vertical mixing was the generic length scale (GLS) tuned to behave as a k- $\varepsilon$  (Warner et al., 2005). Horizontal 145 harmonic mixing of momentum was defined with constant values of 5  $m^2 \cdot s^{-1}$ . 146

The NPZD numerical model coupled with the ROMS model includes dissolved inorganic nitrogen, 148 phytoplankton, zooplankton and detritus (Franks 2002). The initial nitrate concentration was taken from 149 field data collected by IRTA between the years 2009-2012 (ACA, 2012), and the initial phytoplankton 150 concentrations were collected from observation data during the year 2019, whereas the initial zooplankton 151 concentration was estimated from the literature (Rico 2015; Powell et al. 2006). The units in which these data were collected were mg·m-3. The NPZD model uses mmol·m-3 units, so a conversion has been made 152 using the mole fraction of Chl a (893.51 g·mol-1) (see Table 2). The rest of input variables for the ROMS-153 154 NPZD model were acquired from Llebot et al. (2010), and are detailed in Appendix B (Table B1), together 155 with the model equations. Short-term simulations (5 days each) were carried out to analyse the response of 156 biological variables to the wind. This simulation length exemplifies the typical wind events in the area, lasting from 3 to 5 days (except the daily sea breeze during spring and summer) (Ràfols et al. 2017). Six 157 158 experiments were designed with varying wind intensity and direction, and varying freshwater input from 159 channels. The wind parameters are based on wind measurements in the Fangar area (F-Pedrera Balsells et 160 al., 2020a), with weaker down-bay winds (associated to daily sea breeze, DW6 simulation), SE down-bay 161 winds (DW8 simulation), NW up-bay winds (UW10 simulation) and strong NW up-bay winds (UW12 162 simulation) and in addition to a simulation where the flow out of the drainage channels was reduced by half (UW12fr). For theoretical comparison, a simulation was also carried out with  $0 \text{ m} \cdot \text{s}^{-1}$  wind intensity (CALM 163 164 simulation). All simulations are summarized in Table 2. Temperature and salinity conditions were in 165 accordance with those measured within the bay (see field campaign description in F-Pedrera Balsells et al., 166 2021). In addition, Figure 3 shows the horizontal distribution of modelled salinity based on initial 167 conditions interpolated from the observation shown in F-Pedrera Balsells et al. (2021). Freshwater 168 contributions were activated to monitor the evolution of nutrient inputs from the irrigation channels. Both 169 channels (BO and IM, Figure 1) provide nutrients that will be presumably dispersed within the bay due to 170 the combined action of currents and wind.

| Simulation | Wind
direction | Intensity
wind
(m·s -1 ) | Channel
flow (m 3 ·s -1
each
channel) | Initial nitrate
concentration
(mmol·m -3 ) | Initial
phytoplankton
biomass
(mmol·m -3 ) | Initial
zooplankton
biomass
(mmol·m -3 ) |
|------------|-------------------|-------------------------------------------|----------------------------------------------------------------------|-------------------------------------------------------------|----------------------------------------------------------------|--------------------------------------------------------------|
| CALM       | -                 | 0                                         | 7.5                                                                  | 2.73                                                        | 0.27                                                           | 0.08                                                         |
| DW6        | Down-
bay wind | 6                                         | 7.5                                                                  | 2.73                                                        | 0.27                                                           | 0.08                                                         |
| UW10       | Up-bay
wind    | 10                                        | 7.5                                                                  | 2.73                                                        | 0.27                                                           | 0.08                                                         |
| DW8        | Down-
bay wind | 8                                         | 7.5                                                                  | 2.73                                                        | 0.27                                                           | 0.08                                                         |
| UW12       | Up-bay
wind    | 12                                        | 7.5                                                                  | 2.73                                                        | 0.27                                                           | 0.08                                                         |
| UW12fr     | Up-bay
wind    | 12                                        | 3.75                                                                 | 2.73                                                        | 0.27                                                           | 0.08                                                         |

**171 Table 2. Summary of the idealized numerical simulations using the ROMS-NPZD model for Fangar Bay.**

To qualitatively compare the numerical modelling results with real observations, satellite images from 176 Sentinel-2, level 1-C, are used. These satellites carry a single optical instrument, the MultiSpectral Imager 177 (MSI), and its swath width (290 km) and high revisit time (10 days at the equator with one satellite and 2-178 3 days at mid-latitudes) support monitoring of Earth's surface changes. Chlorophyll-a concentrations were 179 computed automatically by the Sentinel Application Platform (SNAP) 180 (https://step.esa.int/main/toolboxes/snap/, accessed on 25 February 2021). The MSI sensor has had an 181 atmospheric correction applied to it with a C2RCC processor (Case 2 Regional CoastColour, Brockmann 182 et al., 2016) to obtain the Chl a images. The images correspond to remote sensing obtained after intense 183 wind episodes (see details in F-Pedrera Balsells et al., 2021).

**185 **3. Results**

Four points within the bay were chosen to investigate the temporal evolution of the biological variables 187 obtained from the NPZD model: in the mouth area (M1), in the centre of the bay (M2), in a coastal area in 188 front of the IM discharge point (M3) and in the innermost part of the bay (M4) (Figure 1, Table 1). Both 189 channels (BO and IM, Figure 1) provide nutrients which increase the concentration of phytoplankton 190 biomass within the bay. Figure 2 shows the time series of the numerical simulation in terms of nitrates and 191 phytoplankton at the four control points. The nitrate concentration tends to decrease gently during the 192 simulation, consistent with the increase in phytoplankton biomass and zooplankton. Stratification 193 conditions (CALM and DW6 simulations) show higher phytoplankton biomass concentrations at the 194 surface due to freshwater fluxes. Substantial differences of phytoplankton biomass between surface and 195 bottom layers are evident in M1, where the stratification tends to be stronger in contrast to the shallowest 196 point (M4). The inner point M4 also shows a clear correlation of the wind intensity and the phytoplankton 197 biomass values: as the up-bay wind intensity increases (i.e. UW12, larger than UW10) the phytoplankton 198 biomass also increases. In all cases the numerical simulations suggest large temporal and spatial variability 199 within the bay.